# Measurement techniques of identifying and quantifying hydroxymethanesulfonate (HMS) in aqueous matrix and particulate matter using aerosol mass spectrometry and ion chromatography.

Eleni Dovrou[1], Christopher Y. Lim[2], Manjula R. Canagaratna[3], Jesse H. Kroll[2], Douglas R. Worsnop[3], and Frank N. Keutsch[1,4,5]

[1]John A. Paulson School of Engineering and Applied Sciences, Harvard University, Cambridge, MA 02138, USA
[2]Department of Civil and Environmental Engineering, Massachusetts Institute of Technology, Cambridge, MA 02138, USA
[3]Center for Aerosol and Cloud Chemistry, Aerodyne Research, Inc., Billerica, MA 02138, USA
[4]Department of Chemistry and Chemical Biology, Harvard University, Cambridge, MA 02138, USA
[5]Department of Earth and Planetary Sciences, Harvard University, Cambridge, MA 02138, USA

*Correspondence to*: Eleni Dovrou (edovrou@g.harvard.edu)

**Abstract.** Oxidation of sulfur dioxide ($SO_2$) in the gas phase and in cloud and fog water leads to the formation of sulfate that contributes to ambient particulate matter (PM). For severe haze events with low light conditions, current models underestimate the levels of sulfate formation which occurs exclusively via the oxidation of sulfur dioxide. We show here that measurement techniques commonly used in the field to analyse PM composition can fail to efficiently separate sulfur-containing species resulting in possible misidentification of compounds. Hydroxymethanesulfonate (HMS), a sulfur(IV) species that can be present in fog and cloud water, has been largely neglected in both chemical models and field measurements of PM composition. As HMS is formed without oxidation it represents a pathway for $SO_2$ to contribute to PM under low light conditions. In this work, we evaluate two techniques for specific quantification of HMS and sulfate in PM, Ion Chromatography (IC) and Aerosol Mass Spectrometry (AMS). In cases where the dominant sulfur-containing species are ammonium sulfate or HMS, differences in AMS fragmentation patterns can be used to identify HMS. However, the AMS quantification of HMS in complex ambient mixtures containing multiple inorganic and organic sulfur species is challenging due to the lack of unique organic fragments and variability of fractional contributions of $H_xSO_y^+$ ions as a function of matrix. We describe an improved IC method that provides efficient separation of sulfate and HMS and thus allows identification and quantification of both. The results of this work provide a technical description of the efficiency and limitations of these techniques as well as a method that enables further studies of the contribution and role of S(IV) versus S(VI) species to PM under low light atmospheric conditions.

## 1 Introduction

### 1.1 Sulfur species in cloud and fog water

Hydroxymethanesulfonate (HMS; $HOCH_2SO_3^-$) is the product of the aqueous-phase reaction between dissolved sulfur dioxide ($SO_2$) and formaldehyde (HCHO) and is considered an important compound in cloud and fog water (Munger et al., 1986; Dixon and Aasen, 1999; Whiteaker and Prather, 2003). HMS is very stable at low pH (pH<6) and is resistant towards oxidation by hydrogen peroxide and ozone; however, it can be oxidized by hydroxyl radicals (Kok et al., 1986; Martin et al., 1989; Chapman et al., 1990). The stability of HMS has a strong pH dependence as it dissociates at high pH values. HMS acid is a strong acid, thus it completely dissociates in water, with a second dissociation constant of pKa=10.2 (R1). (Olson and Hoffmann, 1986; Betterton et al., 1988; Olson and Hoffmann, 1989; Warneck, 1989; Möller, 2014)

$$\text{HCHO} + \text{SO}_3^{2-} \rightarrow \text{HCH(O)}^-\text{SO}_3^- \overset{\text{H}^+}{\rightleftharpoons} \text{HOCH}_2\text{SO}_3^- \quad \text{(R1)}$$

HMS formation results in acidification of the cloud droplets and can contribute significantly to aerosol mass and aerosol sulfur concentration at low pH where it is stable (Dixon and Aasen, 1999). HMS can be retained in aerosol particles after cloud evaporation if the pH is greater than 4.

In cloud and fog water, $SO_2$ reacts with water producing bisulfite ($HSO_3^-$), when 3<pH<6, which further dissociates to form sulfite ($SO_3^{2-}$) when pH>6. Bisulfite and sulfite can be oxidized rapidly by several species such as the hydroxyl radical (OH), ozone ($O_3$), oxygen ($O_2$) and hydrogen peroxide ($H_2O_2$) (Hegg and Hobbs, 1982; Lind et al., 1987; Shen et al., 2012), thus S(IV) species are not expected in PM in significant amounts. Formation of HMS is favourable at high levels of sulfur dioxide and formaldehyde, low levels of oxidants like OH, $H_2O_2$ and $O_3$ (Hegg and Hobbs, 1982; Lind et al., 1987), and cloud and fog pH in the range of

approximately 4-6 (Munger et al., 1984; Munger et al., 1986). Oxidation of dissolved sulfur dioxide by $O_3$ is significant for pH values greater than 4 and oxidation by $H_2O_2$ is considered to be the dominant pathway for the formation of sulfate in cloud and fog water. During haze events oxidant concentrations have been reported to be low resulting in low oxidation rates whereas formaldehyde and sulfur dioxide concentrations have been reported to be high (Ji et al., 2014; Rao et al., 2016; Wang et al., 2016). Therefore, the formation of HMS is favourable under these conditions.

Model simulations under low light conditions in regions with slow photochemistry, such as polluted cities in China and India, underestimate sulfate ($SO_4^{2-}$) concentrations measured in the field using ion chromatography (IC) (Wang et al., 2016), indicating that there is either a missing source of $SO_4^{2-}$ in the model or other sulfur-containing species are misidentified as $SO_4^{2-}$ by IC. During 2009 and 2010 two field campaigns were conducted in Germany (Scheinhardt et al., 2014) reporting the presence of HMS

in particles produced in urban areas. HMS concentrations were highest during winter time in particles with 0.42-1.2 μm diameter size range, although concentrations were low, most likely as not all conditions conductive to HMS formation were met, i.e., there were low light conditions but also low formaldehyde and $SO_2$ concentrations. In January 2013 an extreme winter haze event was recorded over Northern China which resulted in high levels of sulfate measured by IC compared to periods observed before and after the event. The GEOS-Chem chemical transport model (GEOS-Chem CTM) was not able to reproduce the observed $SO_4^{2-}$

concentrations during the haze events despite good performance during other periods, as it under-predicted $SO_4^{2-}$ concentrations by a factor of 4 during the haze periods. Specifically, the model estimated $SO_4^{2-}$ concentrations to be similar for haze and non-haze periods. This suggests that there might be a significant, missing source of $SO_4^{2-}$ (Wang et al., 2014). Wang at al. (2014) suggested that a new heterogeneous pathway of $SO_4^{2-}$ formation could explain the missing $SO_4^{2-}$. Moch et al. (2018) suggested the contribution of HMS to explain the high observed $SO_4^{2-}$ concentrations during these low light haze events with slow

photochemistry. In order to distinguish the two hypotheses, i.e., condensed-phase reactions producing sulfate or contribution from HMS, measurement techniques that allow quantitative speciated measurements of HMS and sulfate are needed.

Measurement of sulfate in ambient PM is common, whereas measurements of HMS have mainly been conducted for fog and cloud water. Studies reporting the presence of HMS in ambient PM using single-particle mass spectrometry have also been conducted

(Neubauer et al., 1996; Neubauer et al., 1997; Whiteaker and Prather, 2003; Lee et al., 2003; Dall'Osto et al., 2009). Two main methods have been used, ion chromatography (IC) and mass spectrometry (MS). For IC a characteristic elution time is used for identification of different ions, including sulfate. For MS the detailed mass spectrum, especially differences in fragmentation patterns, can provide a means to differentiate, in this case, different sulfur-containing species. Moreover, for MS, cations can be observed simultaneously in addition to sulfur-containing ions, whereas for IC a specified IC column with high sensitivity for sulfur-

containing ions has to be used to identify them. In order to distinguish HMS from sulfate using IC or MS, the elution times or the mass spectra and fragmentation patterns, respectively, have to be distinct. (Munger et al., 1986; Chapman et al., 1990; Neubauer et al., 1996; Neubauer et al., 1997; Dixon and Aasen, 1999; Zuo and Chen, 2003; Lee et al., 2003; Whiteaker and Prather, 2003; Dall'Osto et al., 2009)

Sulfate is traditionally measured using IC, but for measurements of PM little attention has been given to the effect of HMS in PM on sulfate measurements. An IC system with alkanol quaternary ammonium analytical column is widely used to separate the main inorganic ions, i.e. $SO_4^{2-}$, $NO_3^-$, $Cl^-$ and $Br^-$ (Hegg and Hobbs, 1982; Wang et al., 2005; Shen et al., 2012). Single-particle mass spectrometry (SPMS) and the Aerodyne aerosol mass spectrometer (AMS) have been used to detect sulfate (Jimenez, 2003;

Murphy et al., 2006; Ji et al., 2014). SPMS and AMS are used for on-line and off-line analysis. During the on-line analysis ambient air is sampled through an inlet to the instrument. For offline analysis, filters collect particles from ambient air, the collected material is extracted into water and after additional dilution the extracts are atomized for analysis via SPMS or AMS.

A variety of technical methods have been used to detect HMS, mainly IC using specific columns (Munger et al., 1986; Dixon and

Aasen, 1999), reverse-phase ion-pair high performance liquid chromatography (HPLC) (Zuo and Chen, 2003), electrospray ionization-tandem mass spectrometry (ESI-MS) (Chapman et al., 1990) and single-particle mass spectrometry (single-particle analysis by laser mass spectrometry: PALMS, rapid single-particle mass spectrometer: RSMS, aerosol time-of-flight mass spectrometer: ATOFMS) (Neubauer et al., 1996; Neubauer et al., 1997; Lee et al., 2003; Whiteaker and Prather, 2003; Dall'Osto et al., 2009). In this work we present an IC method specifically developed to identify and quantify HMS and we discuss the ability

of the AMS to identify and quantify HMS in the presence of sulfate and different cations, to evaluate the matrix effects, under laboratory conditions. In addition, we compare these methods with the technical methods used in previous work.

## 1.2 Previous work identifying HMS using Single-particle Mass Spectrometry, Capillary Electrophoresis and reverse-phase HPLC

Mass spectrometry has been used in the past to identify HMS. Chapman et al. (1990) reported its identification by using an electrospray ionization mass spectrometer (ESI-MS). The characteristic $m/z$ ratio was determined to be $m/z$=111 ($HOCH_2SO_3^-$); to determine a distinct dissociation pattern for HMS the collision-induced dissociation spectrum showed that the $m/z$=80 ($SO_3^-$) and $m/z$=81 ($HSO_3^-$) can be used as characteristic fragment ions for HMS detection. To quantify HMS, $m/z$=81 ($HSO_3^-$) was used as its corresponding peak was larger than the $m/z$=80 ($SO_3^-$) and it showed linear relationship between concentration and ion signal in

the ESI-MS. However, this method may result in noisy spectra for concentrations below 1 ppb, and as discussed later $m/z$=81 ($HSO_3^-$) is not specific to HMS, but rather requires use of tandem mass spectrometry. Chapman et al. (1990) conducted an exploratory study reporting that the quantitative detection limit for HMS can be on the order of 100 $\mu g \cdot m^{-3}$, for typical sampling conditions, using an ESI-MS. Since 1990 there have been advances in the ESI-MS technology that likely result in lower detection limits. However, to our knowledge, these technological changes have not yet provided quantitative evidence of lower detection

limits with respect to HMS analysis.

Neubauer et al. (1996, 1997) explored the possibility of separating sulfur-species, including HMS, by the use of rapid single-particle mass spectrometer (RSMS) in aerosols. Particles are vaporized and ionized by a pulsed laser (248 nm) and analysis is completed by a reflectron time-of-flight mass analyzer. In contrast to ESI-MS, RSMS did not show an $m/z$ ratio of 111

($HOCH_2SO_3^-$) and the dominant signal was $m/z$=64 ($SO_2^+$) when dry particles were analysed. The $m/z$=111 ($HOCH_2SO_3^-$) ion was observed only in the case of acidic aqueous particles. The single particle mass spectrometer provides a wider dynamic range and shorter analysis time compared to ESI-MS however the quantification can be challenging in aqueous matrices due to interference from compounds, such as $(NH_4)_2SO_4$ and methyl sulfonic acid (MSA), present in the sample (Neubauer et al., 1996, Neubauer et

al., 1997). Whiteaker and Prather (2003) used a single-particle aerosol time-of-flight mass spectrometer (ATOFMS), with desorption/ionization laser at 266 nm, to identify HMS in ambient particles and droplets as a tracer for fog processing. In that work, even though the $m/z$=111 ion was observed in some cases when HMS sodium salt was mixed with ammonium sulfate, the identification of HMS in ambient samples was difficult and resulted in uncertainties in the quantification (Whiteaker and Prather, 2003). During a fog event in London, Dall'Osto et al. (2009) also reported the presence of HMS using an ATOFMS. The $m/z$=111

($HOCH_2SO_3^-$) and $m/z$=81 ($HSO_3^-$) ions were identified as markers of HMS. Gilardoni et al. (2016) provided the spectrum of HMS using standard samples. During that study HMS was used as a tracer of aqueous chemistry. Single-particle mass spectrometers have been optimized to overcome sensitivity issues by improving the inlet design, reducing the pump configuration, applying a dual-polarity grid-less reflection design and removing secondary coating of aerosols prior to the analysis (Pratt et al., 2009; Hatch et al., 2014). Such changes can result in higher sensitivity (Pratt et al., 2009; Pratt and Prather, 2012; Hatch et al., 2014). The effect

of these optimizations on the sensitivity to HMS has not been reported.

Lee et al. (2003) conducted a field campaign measuring the chemical composition of aerosols with 0.35-2.5 µM diameter during the 1999 Atlanta Supersite Project. Using a PALMS instrument they identified HMS via the $m/z$=111 ($HOCH_2SO_3^-$) ion. Methylsulfate ($CH_3OSO_3^-$) was also identified by the $m/z$=111 ion, however the authors concluded that due to the low acid

concentrations in the particles and high temperatures in Atlanta, the $m/z$=111 ion could not be assigned to methylsulfate (Lee et al., 2003). Although Song et al (2019) stated that the detection limit of AMS and SPMS for HMS could possibly be lower than the concentration reported by Chapman et al. (1990), 100 $\mu g \cdot m^{-3}$ using ESI-MS, such lower levels of HMS were not able to be detected using these methods. In their study, Song at al. (2019) were able to identify HMS as a component of SOA but they could not quantify it, likely for the reasons outlined below in this work.


Overall, quantification of HMS using single-particle MS methods is challenging due to matrix effects in ambient samples (Neubauer et al., 1996; Neubauer et al., 1997; Whiteaker and Prather, 2003). The sensitivity challenges of these methods with respect to HMS quantification yield the necessity of further study. Aerosol mass spectroscopy (AMS) was used in this work to investigate the ability to identify and quantify HMS and will be described in detail below. However, all mass spectrometry

techniques share the challenge that the majority of the fragments, such as $SO_3^-$ and $HSO_3^-$, are common to different sulfur-containing species, including organic compounds potentially in the measured PM(Ge et al., 2012; Canagaratna et al., 2015; Gilardoni et al., 2016; Song et al., 2019), and that the ratios can depend on other compounds present in PM, such as ammonium and other cations.

Scheinhardt et al. (2014) provided evidence of identification of HMS during two field campaigns conducted in nine sites in Germany using capillary electrophoresis (CE). CE achieved efficient separation of HMS from other compounds using a voltage of -30 kV and hydrodynamic sample injection with 750 mbars was applied. Quantification was achieved through indirect UV detection at 260 nm wavelength and a measurement rate of 20 Hz. The detection limit of HMS was reported as 6-7 $ng \cdot m^{-3}$ and concentrations above this were observed during winter time. The method resulted in successful quantification of HMS in

concentrations $\geq$ 18-21 $ng \cdot m^{-3}$. Concentrations in the range of 6-18 $ng \cdot m^{-3}$ were reported, however this range was

characterized as less reliable in the study. (Scheinhardt et al., 2014) Dabek-Zlotorzynska et al. (2002 and 2005) analysed urban atmospheric aerosol and vehicle emitted samples using CE and reported the presence of HMS. Identification and quantification were achieved using two injection modes, pressure and electrokinetic injection modes, for CE and the detection limits of HMS reported were 0.4 $\mu$M and 0.02 $\mu$M, respectively. The wavelength of the UV detector for indirect detection was at 214 nm. (Dabek-Zlotorzynska et al., 2002) CE may be a method that is highly suitable for detection of HMS and it has high sensitivity and the eluent pH also should prevent decomposition of HMS.

Reverse-phase ion-pair HPLC has successfully been used to separate sulfur-species (Zuo and Chen, 2003). A cetylpyridium-coated $C_{18}$ column was used for efficient separation of the sulfur-spieces and the detection was achieved by indirect UV light absorption. Zuo and Chen (2003) reported the separation and quantification of sulfite, sulfate and HMS at the concentration range of 19-430 $\mu$M, 6.7-430 $\mu$M and 3.8-430 $\mu$M, respectively. This work provides evidence that ion-exchange chromatography can be an efficient method for separation of sulfur-species. Even though mass spectrometry has been widely used for analysis of sulfur-species (Neubauer et al., 1996; Neubauer et al., 1997; Whiteaker and Prather, 2003), there is indication that chromatography methods could provide efficient separation of these species (Zuo and Chen, 2003).

## 2 Experimental

### 2.1 Chemicals and sample preparation

The sodium salt of HMS (Na-HMS) was purchased from Sigma Aldrich (purity 95%). Sodium sulfate ($Na_2SO_4$) and sodium metabisulfite ($Na_2S_2O_5$) were purchased from Sigma Aldrich (purity $\geq$99% in both cases) and used to prepare standards and reference solutions. Sodium metabisulfite was used as a source of bisulfite in the samples as it dissociates rapidly in water to form bisulfite. All solutions were prepared by using filtered Milli-Q water. The samples were analysed at 25 $^{\circ}$C in the pH range of 3 to 12. Six types of samples were prepared to examine all the possible combinations of sulfur-containing species. Solutions containing only sodium sulfate, sodium bisulfite/sulfite, Na-HMS and combinations of Na-HMS with sodium bisulfite/sulfite, Na-HMS with sodium sulfate, and all three sulfur-containing species were prepared and analysed. Hydrogen chloride and sodium hydroxide were used to control the pH of the samples.

### 2.2 Sample analysis

### 2.2.1 Aerosol Mass Spectrometry analysis

The Aerodyne high-resolution time-of-flight aerosol mass spectrometer (HR-ToF-AMS) (DeCarlo et al., 2006) was used to determine the mass spectral signatures of Na-HMS, sodium sulfate and bisulfite. The mass spectra of sodium sulfate, sodium bisulfite and sodium HMS were examined, and the concentration of each solution in the atomizer was 10 mM. The pH of the sample solutions was 6. In addition, solutions containing 20% sulfate and 80% Na-HMS, 40% sulfate and 60% HMS, 60% sulfate and 40% Na-HMS, 80% sulfate and 20% Na-HMS were analysed to evaluate the ability of distinguishing the two species at varying sulfate to Na-HMS ratios. A reference spectrum of ammonium sulfate was also used to investigate the matrix effect. The solutions were atomized by a particle generator (TSI 3076) and subsequently dried before sampled by the AMS. The AMS heater was set in standard operating temperature of 600 $^{\circ}$C. The flow was controlled using an atomizer and the mobility particle diameter was selected at 100.0 nm using an electrostatic classifier (TSI 3082).

**2.2.2 Ion Chromatography analysis**

A Dionex ICS-5000+ Ion Chromatography (IC) system was used to analyse the samples. Two pairs of guard and analytical columns were used. The AG12A-AS12A and the AG22-AS22 pairs (Dionex Ionpac) were selected in order to examine peak separation when columns with different internal coatings (functional groups) are used. The AG22-AS22 column pair was selected due to the fast analysis of inorganic ions that it provides and its general use for main inorganic anion analysis; it is a standard column used for measurement of anions in PM via IC. In addition, the AG12A-AS12A column pair was selected due to its ability to efficiently separate sulfur species. Both column pairs were selected because of the functional group of the analytical column, the hydrophobicity and their efficiency compared to other commercially available columns. The mobile phase during the experiments was 4.5 mM:0.8 mM sodium carbonate: sodium bicarbonate with flow rate $1 \ mL \cdot min^{-1}$. The column and compartment temperatures were both 25°C and the delivery speed and delivery sample volume for the analysis were 4 mL/min and 4 mL. The sample analysis time was 30 min with HMS, bisulfite and sulfate having retention times in the range of 14-16 min.

# 3 Results and discussion

## 3.1 AMS spectra

Samples of sodium bisulfite, sulfate and Na-HMS were analysed individually using the HR-ToF-AMS in order to determine the mass spectral signatures of these compounds (Fig. 1). For Na-HMS organic ions $CHO^+$ ($m/z$=29.00) and $CH_2O^+$ ($m/z$=30.01) are observed. However, these organic ions are observed from many organic species (Canagaratna et al., 2015) and are not specific signatures of HMS. In contrast, methanesulfonic acid (MSA) has been shown to have unique marker ions that contain carbon and sulfur, such as $CH_3SO_2^+$ ($m/z$=78.99) and $CH_3SO_3^+$ ($m/z$=94.98) (Phinney et al., 2006; Huang et al., 2016; Chen et al., 2019). The unique fragmentation of MSA is attributed to the carbon-sulfur (C-S) bond. Chen et al. (2019) also reported that a variety of organic sulfate-containing compounds, that have a C-S bond can be distinguished from inorganic sulfate-containing compounds using AMS due to differences in the fragmentation patterns. In contrast, HMS has a sulfur-carbon-oxygen (S-C-O) bond pattern resulting in lower stability of the molecule. The C-S bond of MSA can be retained after ionization whereas the S-C-O bonds of HMS fragment either from desorption or ionization resulting in the unique marker ions of MSA and lack of specific ions for HMS.

The dominant sulfur-containing $H_xSO_y^+$ ions observed for all samples used in this study were $SO^+$ ($m/z$=47.97) and $SO_2^+$ ($m/z$=63.96). Other weaker ions observed in some of the samples include $SO_3^+$ ($m/z$=79.96), $HSO_3^+$ ($m/z$=80.96) and $H_2SO_4^+$ ($m/z$=97.97). The fractional contributions of each of these ions relative to the sum of all the $H_xSO_y^+$ is shown in Table 1. The $m/z$=111 ($HOCH_2SO_3^-$), which has been previously assigned as the characteristic parent ion of HMS, is not observed in the AMS spectra due to fragmentation from electron-impact ionization and/or thermal decomposition. As shown in Table 1 and Figure 1, the difference between HMS spectra and those from the other species is the absence of signals corresponding to $SO_3^+$, $HSO_3^+$ and $H_2SO_4^+$ for HMS, which are minor fragment ions for the other species. In previous work the fractional contributions of $SO^+$ and $SO_2^+$ ions have been used as indicators of the presence of HMS in ambient samples (Ge et al., 2012; Gilardoni et al., 2016; Song et al., 2019). However, as shown in Figure 1, $SO^+$ and $SO_2^+$ ions are also the two major fragments of the other species (sodium bisulfite, sodium sulfate and ammonium sulfate), and thus their presence and fractional contributions cannot be used as unique indicators for HMS. For example, as shown in Table 1, the fractional contributions of $SO^+$ and $SO_2^+$ ions in Na-HMS, $NaHSO_3$ and $Na_2SO_4$ spectra are very similar, making distinction challenging. Figure 2 and Table 1 demonstrate that the only clear distinction is a minor fragment from $Na_2SO_4$, $SO_3^+$. However, comparison of the mass-spectra of $(NH_4)_2SO_4$ and $Na_2SO_4$ reveal that the

relative intensity of $SO_3^+$ depends strongly on the matrix, in this case the cation, as it is three times as large for $(NH_4)_2SO_4$ compared to $Na_2SO_4$. As seen in Table 1, the fractional contributions of the other $H_xSO_y^+$ fragment ions also depend on the cation. Ammonium sulfate has ion signals at $HSO_3^+$ and $H_2SO_4^+$ that are not present in any of the other species, but Farmer at al. (~~Farmer et al.,~~ 2010) have shown that organosulfate esters such as the trihydroxy sulfate ester of isoprene can also yield $SO_3^+$, $HSO_3^+$ and

$H_2SO_4^+$ ions with relative intensities that are very similar to those observed in ammonium sulfate. In summary, these results show that the lack of truly unique fragments in the HMS spectrum makes identification and quantification of HMS concentrations from AMS spectra challenging, at least when analysing complex ambient samples that contain interfering sulfur-species such as inorganic sulfates, organic sulfates and inorganic bisulfite species. The most accurate quantification of HMS concentrations is likely to be derived from samples that are dominated by HMS. Chen et al. (2019) reported the difficulty in distinction of sulfur-

species due to similarities in fragmentation patterns, which supports the conclusion of this work. The detection of different sulfur organic compounds with AMS is challenging as the fragmentation patterns only have subtle differences and are sensitive to matrix effects.

**3.2 IC chromatographs**

The AG22-AS22 column pair was used to examine the ability to separate HMS and sulfate as well as bisulfite/sulfite and sulfate ions. The AS22 analytical column has the same functional group, alkanol quaternary ammonium, as columns used in previous work for identification of HMS and for ambient analysis during haze events (Munger et al., 1986; Dixon and Aasen, 1999; Wang et al., 2005; Cao et al., 2014; Cheng et al., 2016). The AS22 analytical column provides a direct comparison to this class of columns. The analytical columns can also be classified with respect to the eluent. The types of columns used in previous studies were the

Dionex Ionpac AS11, AS11-HC and AS4A where the AS11 and AS11-HC are anion hydroxide columns and the AS4A is an anion carbonate column. The AS22 analytical column (diameter=4 mm and length=250 mm of the column) is also classified as an anion carbonate column. Anion hydroxide columns are columns that require a strong base eluent to maintain their sensitivity. In contrast, anion carbonate columns need a neutral eluent.

Six samples containing either only sulfate, bisulfite/sulfite, HMS, combination of HMS with bisulfite/sulfite, HMS with sulfate and all three sulfur-containing species were analysed using the AG22-AS22 column pair in a pH range of 3-12. At pH 3-6 the dissolved sulfur dioxide will be in the form of bisulfite ($HSO_3^-$) and at pH>6 it will be in the form of sulfite ($SO_3^{2-}$). The three pH values of solution examined were pH=3, 6 and 12 whereas the eluent pH was ~7. In all cases sulfate and HMS or sulfate and bisulfite/sulfite were not clearly separated (Fig. 3, a and b). In addition, HMS and bisulfite/sulfite had the same retention time

indicating that their separation is not possible in this system. Each sample analysis was conducted 4 times with individual sample preparation before each analysis. The area of the peaks was almost identical for sulfate and HMS in all 4 runs, with a difference only of 0.06 and 0.08 mM, respectively.

In order to examine the possibility of separating sulfate and HMS we used the AG12A-AS12A column pair. The AS12A analytical

column has an alkyl quaternary ammonium functional group. The AS12A analytical column (diameter=4 mm and length=200 mm of the column) is an anion carbonate column, with respect to the eluent, used to analyse inorganic compounds and has the ability to separate sulfur species. The same samples were analysed under the same conditions and the column achieved efficient separation of sulfate and HMS and also sulfate and bisulfite/sulfite (Fig. 3, c and d). HMS and bisulfite/sulfite were not able to be separated

as they had the same retention time in this case as well (Fig. 4). The efficiency and the clear separation of the peaks that the column provides allows for quantification of HMS when bisulfite/sulfite are not present.

Calibration standards were prepared and analysed to determine the retention times (Fig. 5). Each sample was a single component sample containing only one of the sulfur-species. The detection limit of sulfate and HMS was experimentally determined as 0.2 µM and 0.8 µM. The equivalent $ng \cdot m^{-3}$, assuming filter collection of ambient samples with sampling rate of ~80 L · min, sampling time of ~6 hr and extraction volume of 20 mL, are ~13 $ng \cdot m^{-3}$ and ~62 $ng \cdot m^{-3}$. The detection limits were determined by conducting sample runs of different concentrations. The concentration, C, for which the IC could not provide a clear peak was identified and sample runs were conducted for concentrations C+n, where n=0.2 mM. The concentration for which the baseline and the peak were clearly distinguishable was defined and 6 runs were conducted for this specific concentration to verify it. The standard deviation of that concentration was estimated. Blank samples were analysed for each compound and the mean value and the standard deviation was determined. Considering 99% confidence interval the limit of blank was calculated as the mean blank value plus the product of the standard deviation of the blank and 2.58, value which corresponds to 99% confidence level (Limit of blank = (mean of bank) + 2.58·(standard deviation of blank)). The detection limit was estimated as the sum of the limit of blank and the product of the standard deviation of the low concentration and 2.58 (Detection limit = (Limit of bank) + 2.58·(standard deviation of low concentration)). Standards were prepared before each experiment to ensure their stability and avoid possible decomposition if stored for a prolonged period of time. The retention time of sulfate was 14.2-15.2 min for the system with the AS22 column and 10.8-11.2 min for the system with the AS12A column. The retention time of HMS was 14.8-15.2 min and 8.8-9.2 min, respectively. Interestingly, for the HMS and bisulfite individual samples a small amount of sulfate was produced, corresponding to 0.4% of the total signal due to oxidation from oxygen.

Comparing the results from the two column pairs, it was determined that for the AS22 analytical column the HMS peak appears slightly after the sulfate peak whereas for the AS12A analytical column the HMS peak appears before the sulfate peak. Using the AS12A analytical column, sulfate represents 55.2% of the total area signal and HMS 44.8% when a sample of 2 mM of HMS and 2 mM of sulfate was analysed. In contrast, for the AS22 analytical column the area signal of sulfate was 63.6% and HMS was 31.8% for both pH=3 and 6. The peaks were connected and there was no baseline separation thus the software automatically separated the peaks by a vertical line at the minimum point between them. The software allows for determination of the baseline which could result in quantification of the compounds by elevating the baseline to the minimum point between the connected peaks and disregarding the area below. When this was applied a significant underestimation of the concentration, ≳15% of HMS with 4% uncertainty, of the compounds was observed, therefore the software automatic separation was selected to be used. To be more specific, 4% uncertainty corresponds to the concentrations measured in multiple runs, thus the precision of the HMS concentration measurements, and ≳15% underestimation is the underestimation in HMS concentration when a sample containing both sulfate and HMS is analysed and compared with the HMS concentration of a sample containing only HMS. Therefore, the percent underestimation shows the lack of accuracy of the measurements when these two sample types are analysed with the AS22 column. The percentages of HMS and sulfate were obtained considering the software separation of the peaks and the underestimation was determined by obtaining the calibration curves for sulfate and HMS and examining known concentrations. If the concentrations are at lower levels, corresponding to ≲30 µM of HMS, value experimentally estimated under laboratory conditions, which is equivalent to ≲2 $µg \cdot m^{-3}$, assuming filter collection of ambient samples with sampling rate of ~80 L · min, sampling time of ~6 hr and extraction volume of 20 mL, an aliquot of which, 4mL, is used for sample analysis, and sulfate is of equal or higher concentration, the peaks corresponding to HMS and sulfate have lower area signals and will be treated as one peak. For pH=12 the

peaks could not be distinguished. Therefore, when the AS22 analytical column was used the sulfate area signal was increased by 8.4% and the HMS area signal was decreased by 13% compared to the case of the AS12A column was used.

Considering the intensity of HMS and sulfate for the AS12A in the mixed sample, the intensity of the sulfate and HMS peaks was 26.2 μS·min and 21.3 μS·min, respectively, which is the same when HMS and sulfate samples were analysed individually. In contrast, in the case of the AS22, the intensity of the HMS and sulfate peaks was 13.7 μS·min and 30.2 μS·min, respectively. However, when samples containing only HMS and only sulfate were analysed the intensity was 9.3 μS·min and 33.9 μS·min, respectively. Thus, the intensity of the peak of HMS in the sample that contained both HMS and sulfate was 4.4 μS·min higher compared to the sample that had only HMS. The sulfate peak intensity was 3.7 μS·min lower in the sample that contained both HMS and sulfate compared to the sample that had only sulfate. Thus, the area signal of the sulfate increased but the intensity of the peak was decreased, and the reverse phenomenon was observed for HMS. Considering both the signal contribution and the intensity of the compounds, the results indicate that amounts of both compounds are probably incorporated in both peaks and since we have an increase in the area of sulfate it is more likely that some of HMS is attributed to sulfate in this analysis.

The AS22 and AS12A columns have different technical characteristics (Table 2). The difference in the retention times is due to the functional groups (internal coating) of the columns and thus their ability to separate ions. Sulfate is more polar than bisulfite/sulfite, therefore it is expected to have a stronger binding on the stationary phase (functional group) which results in a longer retention time. HMS and bisulfite/sulfite are not separated as they have very similar polarity. In addition, the AS22 analytical column is longer than the AS12A analytical column, which affects the retention time of the examined compounds. The eluent is also a technical aspect that differs between the two columns. The AS12A is an anion carbonate column, thus the eluent is neutral with respect to the pH, whereas the AS22 column is an anion hydroxide column, thus the eluent is basic with respect to pH. The stability of HMS has a strong pH dependence as it dissociates at high pH values. The stability of HMS has a strong pH dependence as it dissociates at high pH values. Therefore, the use of a neutral pH eluent allows to avoid HMS decomposition during analysis. The majority of columns with alkyl quaternary ammonium functional group require neutral pH eluent, which results in efficient separation of sulfur species.

Another factor that can affect the retention time of the compounds is the hydrophobicity of the stationary phase of the column. The AS22 analytical column has ultralow hydrophobicity whereas the AS12A analytical column has medium hydrophobicity resulting in more efficient separation of species within a single family. An ultralow hydrophobicity results in faster retention for non-polar compounds and will cause polar substances of the matrix to accumulate in the column, possible leading to undesirable effects such as misidentification of compounds and shifted retention times. Non-polar compounds will be transferred down the column more readily whereas polar compounds, such as sulfate and bisulfite/sulfite, might not be eluted efficiently by the eluent resulting in unrealistic retention times and peak shapes in the chromatograph. This factor can possibly explain the longer retention time of HMS compared to sulfate when the AS22 column is used, as sulfate has higher polarity than HMS.

## 4 Conclusion

This study investigates techniques used to identify and quantify HMS and sulfate in PM that contains both species. Two main methods were examined, IC and AMS. HMS and sulfate can be efficiently separated and quantified using an IC system with an analytical column that has an alkyl quaternary ammonium functional group (i.e. AS12A). However, using a column with alkanol

quaternary ammonium functional groups (i.e. AS22) quantification of sulfate and HMS is challenging as the peaks are not separated efficiently and they may be identified as one species, typically sulfate. Hence, HMS could possibly be mistaken as sulfate in field measurements. Using an IC system, the detection limit of quantifying HMS and sulfate is 0.8 μM and 0.2 μM, respectively, and the required concentration needed to distinguish HMS and sulfate was determined to be >30 μM of HMS and the sulfate concentration has to be lower concentration than that of HMS. These sulfur-species can also be distinguished using a variety of mass spectrometry instrumentation if the HMS concentration is high compared to that of other sulfur species present in the analysed sample. However, the fragments that are used for HMS quantification are common to other sulfur-species and are subject to interference from organosulfates and inorganic sulfates. Moreover, this interference can vary with the matrix, in particular cations present in the sample (i.e. $Na_2SO_4$ versus $(NH_4)_2SO_4$).

The results obtained in this study may help explain the case of January 2013 haze event in Northern China (Wang et al., 2014) where models under-predicted sulfate levels compared to observations. During the study of the 2013 haze events, field measurements, analysed using an alkanol quaternary ammonium column, showed 70-90% increased sulfate concentrations compared to the model simulations (Wang at al., 2014), and one explanation that has been proposed is that HMS was quantified as sulfate. Similarly, AMS measurements may have identified HMS as sulfate as explained above. This is also consistent with the explanation provided by Moch at al. (2018) and Song at al. (2019).

Applications of both IC and AMS methods to ambient samples from similar conditions of the January 2013 haze event in the future will provide an opportunity to characterize the efficiency of identification and quantification of HMS and sulfate in complex mixtures and the degree to which non-oxidative reactions of $SO_2$ contribute to ambient PM, especially for low light conditions associated with severe haze events. If HMS is not suspected to be present in field samples, it can be overlooked and possibly misidentified as sulfate.

*Author contributions.* FNK initially conceived of the work. ED developed the specific ion chromatography method described in this work, performed the experiments and analysed the data. CYL and ED conducted the aerosol mass spectrometry experiments and CYL, ED and MRC analysed the data. ED prepared the manuscript with contributions from CYL, MRC, JHK, DRW and FNK.

*Competing interests.* The authors declare they have no conflict of interest.

*Acknowledgments.* This work was supported by the Harvard Global Institute. The authors thank J. William Munger for helpful statistical and HMS related discussions, Loretta J. Mickley and Jonathan M. Moch for helpful preliminary discussions.

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

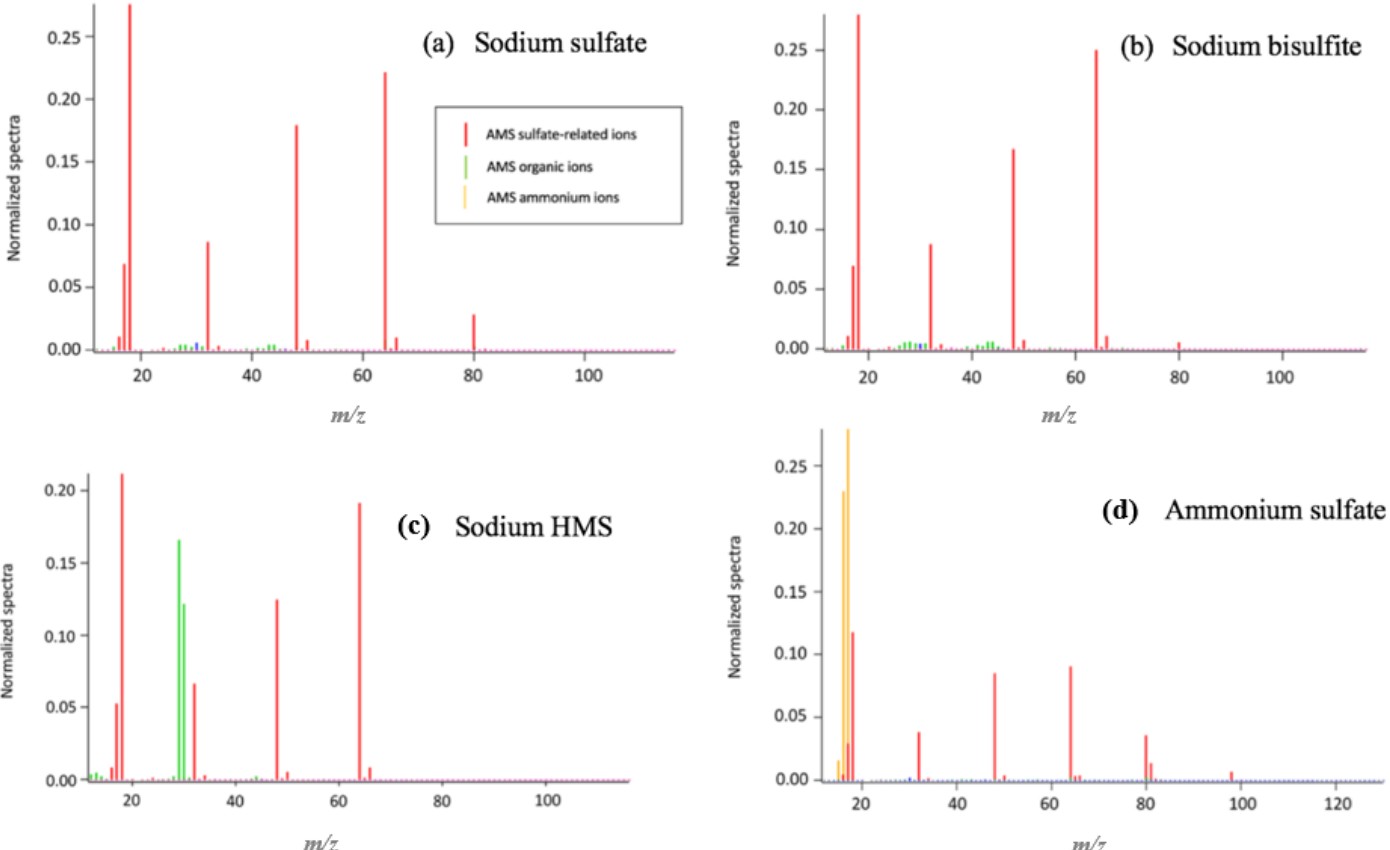

**Figure 1: Samples analysis using the HR-ToF-AMS. (a) Sodium sulfate fragmentation. The main peaks are SO$^+$ (*m/z*=48) and SO$_2^+$**
**(*m/z*=64). (b) Sodium bisulfite fragmentation. The spectrum is similar to the sodium sulfate spectrum indicating that their distinction is**
10 **not possible. (c) Sodium HMS fragmentation. The main differences which allow the distinction among HMS, bisulfite and sulfate is the**
**presence of the organic ions and the absence of the SO$_3^+$ ion (*m/z*=79.96) in the HMS spectrum. (d) Ammonium sulfate fragmentation was**
**used as reference. Similar to (a), (b) and (c) the main ions are SO$^+$ (*m/z*=48) and SO$_2^+$ (*m/z*=64). Ammonium sulfate is also distinguished**
**from HMS due to the presence of the SO$_3^+$ ion (*m/z*=79.96), HSO$_3^+$ ion (*m/z*=80.96) and H$_2$SO$_4^+$ ion (*m/z*=97.97). The pH of all samples was**
**6 and the temperature 25 °C.**

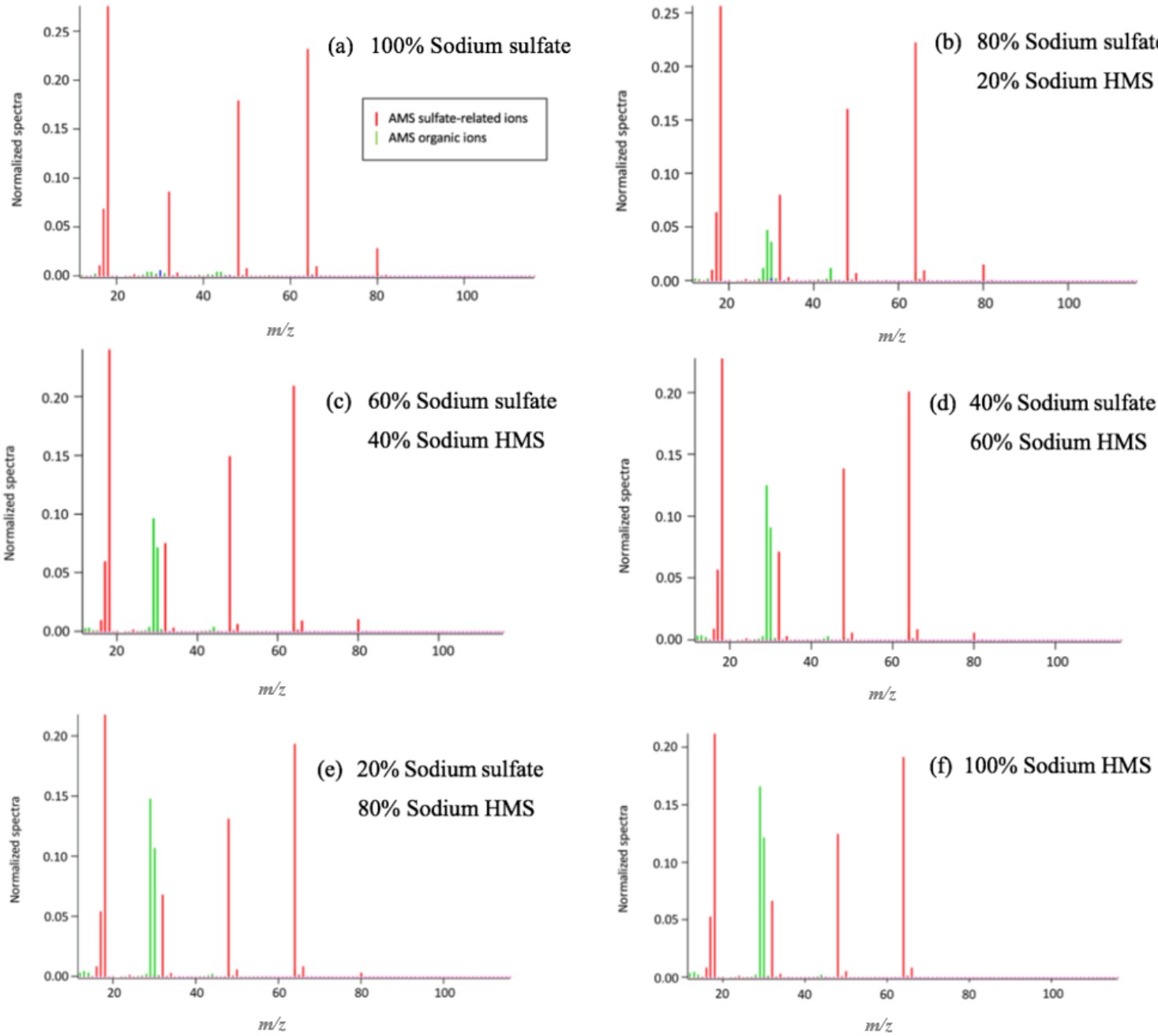

**Figure 2: HR-ToF-AMS analysis of aqueous samples containing sodium sulfate and sodium HMS. (a) 10 mM (concentration in the atomizer) of sodium sulfate was analysed to obtain its signature based on its fragmentation. (b) A sample containing 80% sodium sulfate and 20% sodium HMS was analysed. (c) The sample was prepared with 60% of sodium sulfate and 40% of HMS. Consecutively, (d) presents the fragmentation of a sample with 40% sodium sulfate and 60% sodium HMS, (e) 20% sodium sulfate and 80% sodium HMS and (f) the fragmentation of 10 mM HMS sample. Increase of the concentration of HMS results in the increase of the organic ions and the decrease of the $SO_3^+$ ion ($m/z$=79.96). The dominant ions, $SO^+$ ($m/z$=47.97) and $SO_2^+$ ($m/z$=63.96), seem to remain constant. The pH of all samples was 6 and the temperature 25 ºC.**

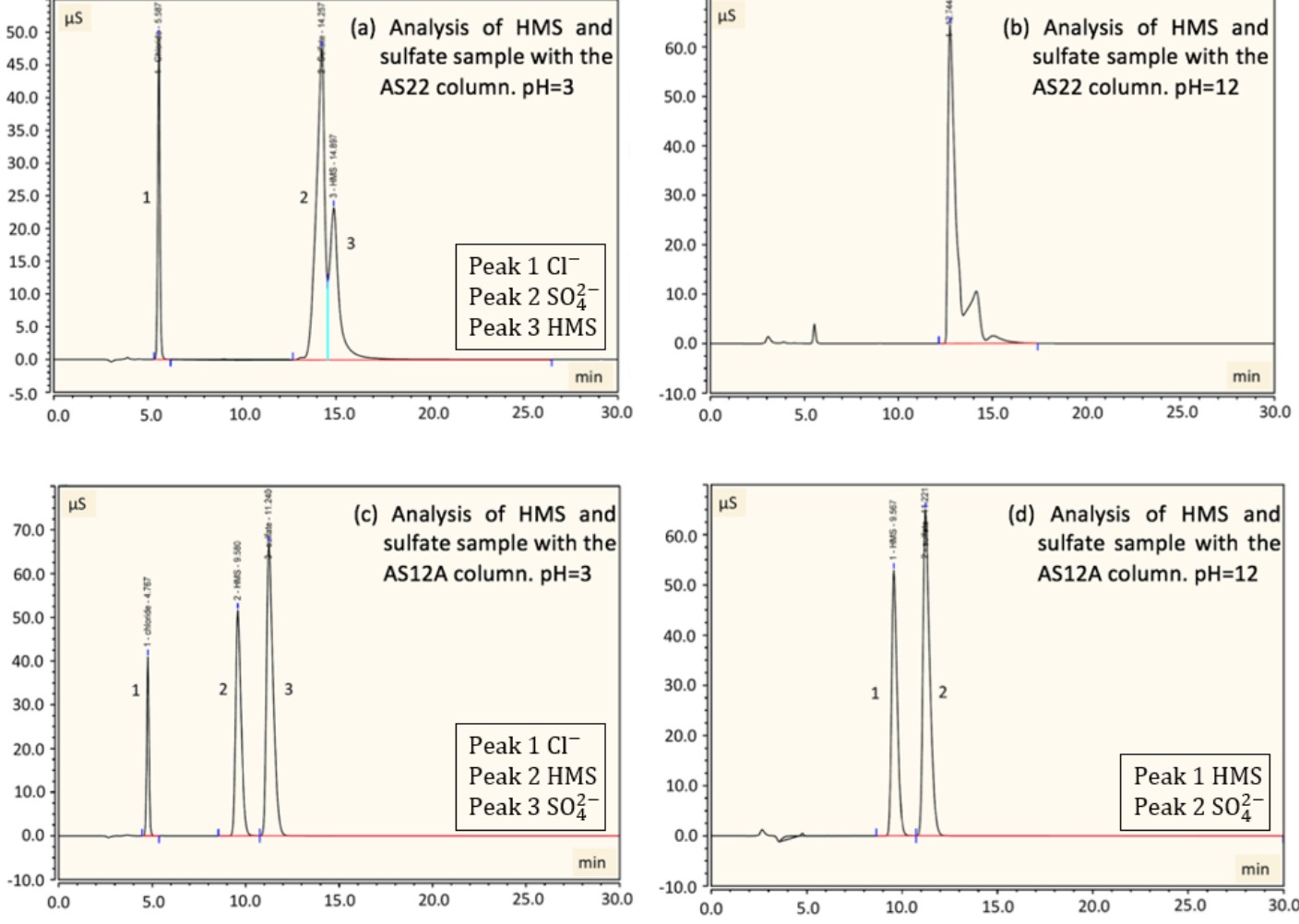

**Figure 3: Detection and separation of sulfate and HMS using two ion chromatography systems. The first system, corresponding to (a) and (b), had an AG22 guard column and AS22 analytical column (alkanol quaternary ammonium functional group) and the second system, corresponding to (c) and (d), had an AG12A guard column and an AS12A analytical column (alkyl quaternary ammonium functional group). (a) A sample of 2 mM of HMS and 2 mM of sulfate at pH=3 was analysed using the AG22-AS22 column pair. Peak 1 represents the chloride at 5.6 min, as HCl was used to acidify the solution, peak 2 represents the sulfate at 14.3 min and peak 3 represents the HMS at 14.9 min. The separation of sulfate and HMS is not efficient. (b) The same analysis was performed at pH=12 indicating that the column fails to provide clear peaks in basic pH. The analysis was repeated using the AG12A-AS12A column pair in acidic (pH=3, (c)) and basic (pH=12 (d)) conditions. (c) Peak 1 represents the chloride at 4.8 min, peak 2 represents the HMS at 9.6 min and peak 3 represents the sulfate at 11.2 min. (d) Peak 1 represents the HMS at 9.6 min and peak 2 represents the sulfate at 11.2 min. The results indicate that the column separates efficiently the two species in both the cases of pH=3 and pH=12.**

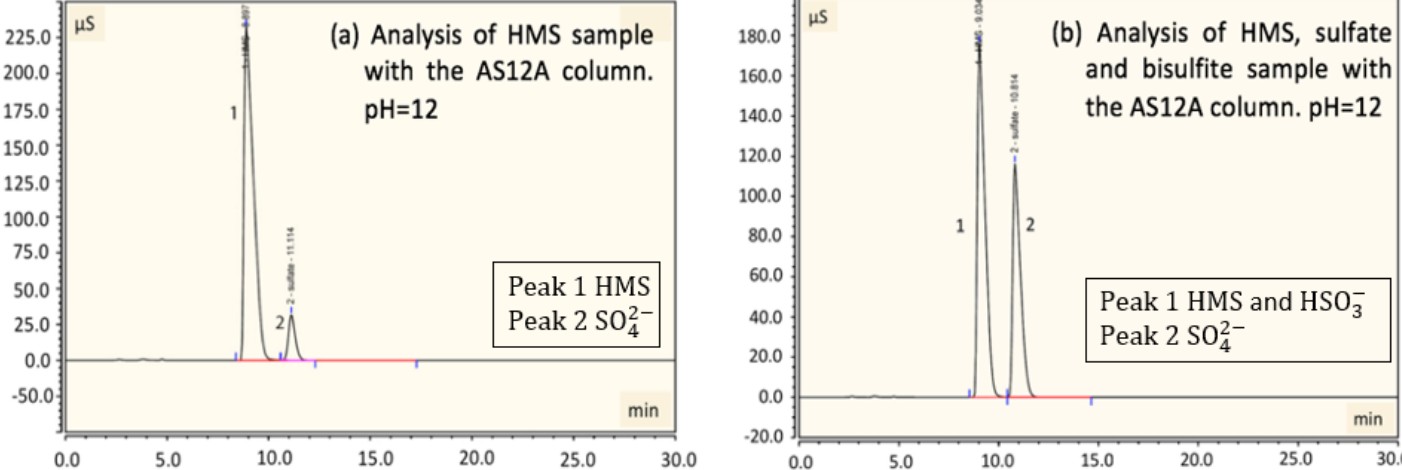

**Figure 4: Detection and separation of HMS and sulfate using an ion chromatography system with AS12A analytical column and AG12A guard column. (a) A sample of 2 mM of HMS at pH=12 was analysed. A small amount of sulfate is produced due to oxidation by oxygen. The column separates efficiently HMS and sulfate. Peak 1 represents the HMS at 9.6 min and peak 2 represents the sulfate at 11.2 min. (b) A sample of 2 mM of HMS, 2 mM of sulfate and 4 mM of bisulfite at pH=12 was analysed. Peak 1 represents the HMS at 9.0 min and peak 2 represents the sulfate at 10.8 min. The separation of sulfate and HMS is efficient; however, separation of bisulfite and HMS was not possible. Samples were examined at pH=3 and 6 as well with similar separation efficiency as the aforementioned samples.**

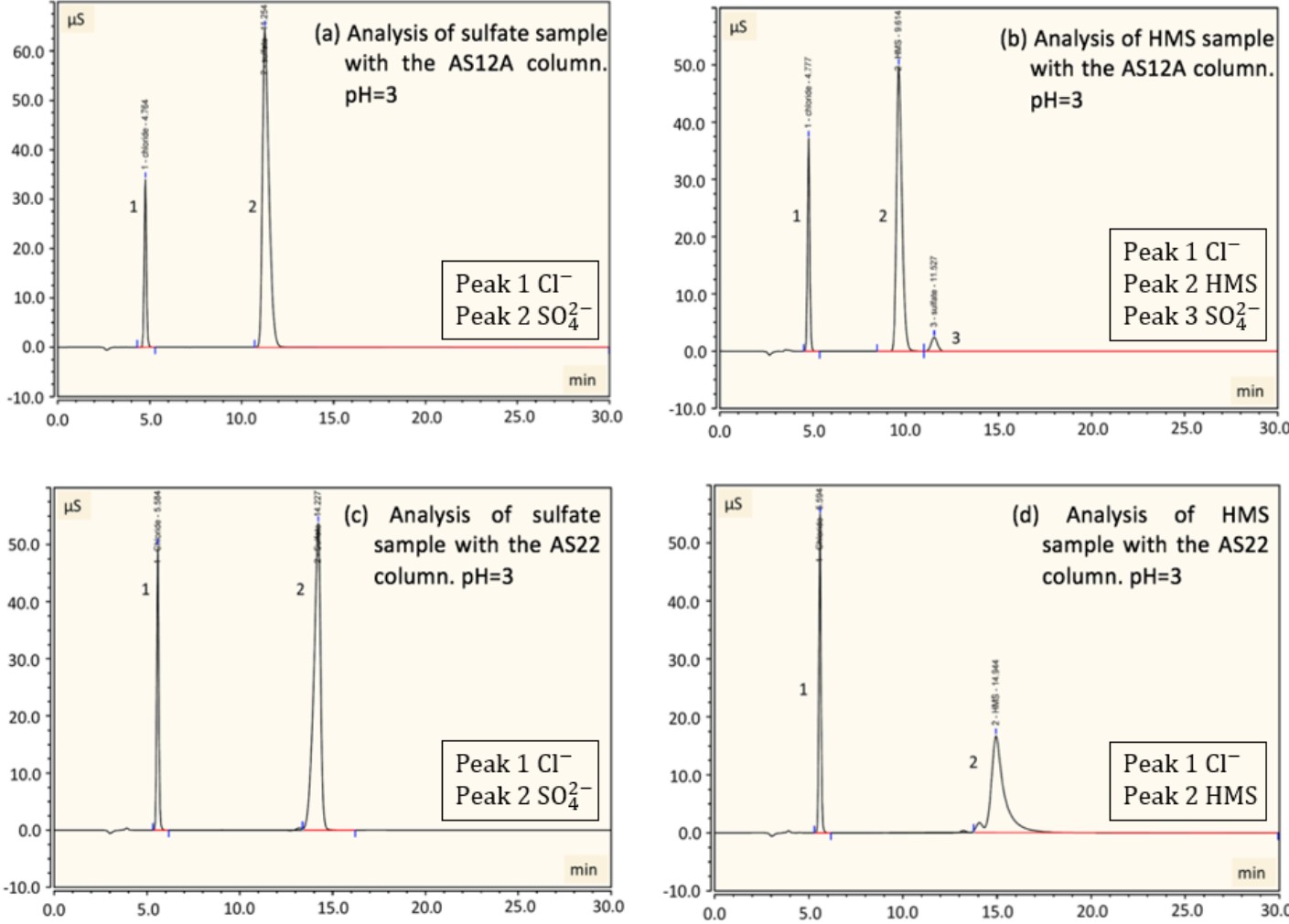

**Figure 5: Sample analysis of sulfate and HMS using two ion chromatography systems. The first system, corresponding to (a) and (b), had an AG12A guard column and an AS12A analytical column (alkyl quaternary ammonium functional group) and the second system, corresponding to (c) and (d), had an AG22 guard column and AS22 analytical column (alkanol quaternary ammonium functional group). The pH was acidic (pH=3) and all samples were in room temperature (25 °C). (a) A sample of 2 mM of sulfate was analysed using the AG12A-AS12A column pair. Peak 1 represents the chloride at 4.8 min, as HCl was used to acidify the solution, and peak 2 represents the sulfate at 11.3 min. (b) A sample of 2 mM of HMS was analysed using the AG12A-AS12A column pair. Similarly, peak 1 represents the chloride at 4.8 min and peak 2 represents the HMS at 9.6 min. Interestingly, a 0.4% of HMS is oxidized by oxygen, resulting on the production of sulfate (peak 3). (c) A sample of 2 mM of sulfate was analysed using the AG22-AS22 column pair. Peak 1 represents the chloride at 5.6 min and peak 2 represents the sulfate at 14.2 min. (d) A sample of 2 mM of HMS was analysed using the AG22-AS22 column pair. Similarly, peak 1 represents the chloride at 5.6 min and peak 2 represents the HMS at 14.9 min. Both systems provide efficient identification of sulfate and the chromatographs represent sulfate with a smooth shaped peak. In addition, both systems identify HMS; however, the system with the AG22-AS22 column pair indicates that the quantification of HMS might not be possible due to the discontinuous shape of the peak.**

**Table 1: Fractional contributions of $SO^+$, $SO_2^+$, $SO_3^+$, $HSO_3^+$ and $H_2SO_4^+$ to the sum of their intensities in AMS spectra.**

| Sample | $SO^+$ fraction (m/z=47.97) | $SO_2^+$ fraction (m/z=63.96) | $SO_3^+$ fraction (m/z=79.96) | $HSO_3^+$ fraction (m/z=80.96) | $H_2SO_4^+$ fraction (m/z=97.97) |
|---|---|---|---|---|---|
| Sodium sulfate ($Na_2SO_4$) | 42% | 56% | 2% | 0% | 0% |
| Sodium bisulfite ($NaHSO_3$) | 38% | 62% | 0% | 0% | 0% |
| Na-HMS | 40% | 60% | 0% | 0% | 0% |
| Ammonium sulfate (($NH_4)_2SO_4$) | 45% | 46% | 6% | 2% | 1% |
| 80% $Na_2SO_4$ and 20% NaHMS | 42% | 57% | 1% | 0% | 0% |
| 60% $Na_2SO_4$ and 40% Na-HMS | 42% | 57% | 1% | 0% | 0% |
| 40% $Na_2SO_4$ and 60% Na-HMS | 35% | 65% | 0% | 0% | 0% |
| 20% $Na_2SO_4$ and 80% Na-HMS | 40% | 60% | 0% | 0% | 0% |

5   **Table 2: Technical characteristics of the columns used for the ion chromatography analysis.**

| Analytical column | Guard column | Functional group | Eluent classification | Analytical column diameter (mm) | Analytical column length (mm) | Hydrophobicity |
|---|---|---|---|---|---|---|
| AS22 | AG22 | Alkanol quaternary ammonium | Anion carbonate | 4 | 250 | Ultralow |
| AS12A | AG12A | Alkyl quaternary ammonium | Anion carbonate | 4 | 200 | Medium |