# Peer review of "Measurement techniques of identifying and quantifying Hydroxymethanesulfonate in cloud water and particulate matter."

_Atmospheric Measurement Techniques, 2019_

## Referee Comment (RC1) · Anonymous Referee #2 · 1 Jun 2019

Dovrou et al describe the laboratory measurement of HMS and sulfate by IC and AMS. A useful aspect of this work is the evaluation of the AMS measurement of HMS in the presence of other sulfur-containing species, which builds upon the work of Gilardoni et al (2016, PNAS), who showed the AMS mass spectrum of HMS only. Line numbers here refer to those in the tracked changes version of the manuscript.

It is unfortunate that the authors decided not to test the AG18-AS18 columns used in the URG AIM-IC. I understand that testing a new column would require additional laboratory work, but I really believe it would significantly extend the usefulness of the manuscript, as the AIM-IC is being used by many researchers in China. In the reviewer response, the authors note that they expect that the columns "will not allow efficient separation of HMS and sulfate", but this is highly qualitative and likely depends on the

eluent and run conditions. I urge the authors to reconsider adding the AG18-AS18 column testing to their work here.

Major Comments:

Title: The goal of this work, as stated through the paper, is to examine methods for the measurement of HMS in PM. Therefore, I suggest that the authors revise the title of their manuscript to specifically mention HMS and PM, rather than fog and cloud water.

Page 7, Line 19 and Page 8, Lines 5-6: The authors state "this method may result in noisy spectra for concentrations below 1 ppb" when discussing ESI-MS, but the paper cited (Chapman et al.) is from 1990. The signal/noise will depend on the mass analyzer used, in addition to the ionization method, and there have been great advances in mass analyzers and associated sensitivities over the past 30 years. Similarly, the LOD for the ESI-MS method is quoted as $\sim$100 ug/m3, but again, I expect this would have changed significantly over the time since publication. Therefore, these statements should be qualified, and rather future work should be motivated here to examine current sensitivities on ESI-MS instruments.

Page 8, Lines 11-12: In reviewing Whiteaker et al (2003, Atmos. Environ.) based on the reviewer response, this paper does not cite a lack of sensitivity by the ATOFMS for detecting HMS, making the statement on Line 11 misleading. Rather, Whiteaker et al. discuss the matrix effects of ammonium and sodium, which impact the peak area detected; I cannot find evidence in this manuscript that the LOD for HMS would be high, indicating a lack of sensitivity, and no comparison is provided to other techniques. Therefore, I suggest the authors remove the phrase "and lack of sensitivity" and instead suggest in the paper that a study of the sensitivity of single-particle mass spectrometry instruments to HMS is an area of future work needed (as eluded to now on Page 7, Lines 37-38).

Page 7, lines 6-8: PALMS is a single-particle mass spectrometry instrument. Please correct here. Please also note that the single-particle mass spec papers listed here do

not represent a complete list, as implied. Either include "e.g." in front of the literature list, or conduct a more thorough literature search. Similarly, Section 1.2 describes each of the single-particle mass spectrometry studies listed here, but again, this is only a subset of published work on the subject, which is not reflected in the summaries presented. I'd encourage the authors to consider in Section 1.2 to conduct a more thorough literature search, and rather than describing each paper one-by-one, include a brief overview/summary of the observations.

Page 11, Lines 12-14: The addition of the IC LODs and explanation of conversion to ambient mass concentration is very useful. However, please clarify how the LODs were determined and what they refer to, as there as multiple methods and definitions used in chromatography for LODs.

Page 11, lines 24-27: I am confused at why a significant underestimation occurred when the elevated baseline was used. How was this determined? Was a calibration curve obtained and then a known concentration run to check? What is "significant" in this case? Please clarify.

Section 3.2: What are the uncertainties in the percentages reported? Is reporting to one decimal place appropriate? Where the samples run in triplicate?

Section 4: Add a statement of the required concentration needed to distinguish HMS and sulfate (discussed on page 11), as this seems like it will significantly impact the recommendation of the necessary mass loading for ambient samples.

Additional Comments:

Section 1.1: While the pivotal work of Munger et al 1986 (Science) is cited later in the manuscript, it would be highly valuable and most appropriate for this to be cited in the first paragraph of Section 1.1, as it sets the stage for the entirety of this work.

Page 2, Line 16-20: Please provide references for these statements.

Page 6, Lines 27-28: Rephrase statement "...measurements of HMS have mainly

been conducted of fog and cloud water only" as there have been many ambient PM measurements of HMS by single-particle mass spectrometry.

Page 6, Lines 30-32: This statement is confusing as written. Please clarify.

Please write out all acronyms used throughout the manuscript (as one example (there are others) – see use of RSMS and ATOFMS on Page 7, Line 7).

Page 7, Lines 29-30: Single-particle mass spectrometers have several lasers. Please correct "operating laser" to "desorption/ionization laser".

Page 7, Line 35: Matrix effects are inherent to the laser desorption/ionization process and have nothing to do with the inlet design, pump configuration, and reflectron. Rather matrix effects are associated with the competition in ion formation. Please correct here.

Section 1.2: This section should describe the previous work of Gilardoni et al (2016, PNAS), who showed the AMS mass spectrum of HMS only. I realize that this paper is cited, but it would be useful for it to be described in the introduction to set the stage for how the current work builds upon this previous work.

Page 7, Line 13: Consider replacing "RSMS, PALMS, AToFMS" with "Single-particle Mass Spectrometry" as these are simply three of many types of single-particle mass spectrometers.

Page 7, Line 29: Correct "AToFMS" to "ATOFMS".

Page 8, Lines 5-6: Please move these sentences the first paragraph of Section 1.2, where ESI-MS is discussed.

Page 8, Lines 6-7: Please clarify this sentence. Song et al. (2018) did detect HMS by SPAMS, even though the opposite seems to be stated in this sentence, with the opposite statement then in the following sentence.

Page 9, Line 17: Can you provide references here for the common use of this column?

Page 10, Line 7: Please clarify "other species" here.

Page 10, Line 14: Fix reference formatting here.

Section 3.2: Some of this section repeats the methods and could be condensed.

Page 13, Line 1: I'm confused by the statement "Applications of both IC and AMS methods to the same ambient samples in the future" as isn't a finding of this work that the AMS is unable to distinguish between HMS and sulfate. Also, I'm confused because I thought these samples were not available for analysis based on the reviewer response. Please clarify.
* * *

---

## Referee Comment (RC2) · Anonymous Referee #3 · 10 Jun 2019

General Ths is an interesting contribution as to the problem to better differentiate between S(IV) and S(VI) species in PM analysis, when performed through AMS measurments.

I feel it is right in the centre of papers of interest for AMT.

Generally, the paper could a bit more reference to available offline analysis work.

I think this paper could be accepted for publication in AMT subject to arevision somewhere between minor and major.

Details

Title: The title as it stands now is very broad. Mayb it could be phrased a bit more

specfific ? Wouldn't it make sense to clearly mention HMS ?

Introduction: It covers quite soome aspects, but at times there could be some more coverage. Maybe the authors can check again, HMS has been discussed a bit more often.

Page 3, section 1.2: I find it starngethat here the very successifully applied CE (capillary electrophoretic) separation and determination is not described. This is a mayor flaw and needs to be corrected. See Scheinhardt et al ., but especially references therein, Kramberger et al.

Page 5, line 29ff : For MSA you should possibly reference Huang, Shan, et al. "Latitudinal and seasonal distribution of particulate MSA over the Atlantic using a validated quantification method with HR-ToF-AMS." Environmental science & technology 51.1 (2016): 418-426.

Page 6, section 3.1.: Maybe it would be good to carry the conclusion of this section into the abstract: It is very difficult if not even impossible to identify or even quantify HMS through AMS only.

Also, the HPLC method presented here does not fully convince. Please give numbers of merit for it and compare to all existing offline analytical techniques. Could you discuss wether AMS paralleled by filter sampling and CE analysis wouldn't be a valuable option ? In this view, the discussion at the end of the paper should be widened.

---

## Referee Comment (RC3) · Anonymous Referee #1 · 11 Jun 2019

In this manuscript, two common sulfate measurement methods, ion chromatography and aerosol mass spectrometry, are evaluated for their ability to measure HMS (hydroxymethanesulfonate) in the presence of sulfate and other organic sulfur particles in atmospheric samples. It also describes an improved IC method that can allow for better chromatographic separation and quantification of HMS. The topic and quality of this works are suitable for publication on AMT. It is recommended for acceptance after the authors respond to the comments outlined below.

First of all, the title of the paper needs to be revised to better represent the main contents of the manuscript. For example, this work focuses exclusively on the AMS and IC methods, although there are other techniques available for HMS quantification. Also, the current title reads as if the method targets a range of sulfur containing compounds,

[Figure]

but in reality it is primarily for HMS.

Secondary, more analytical details on the IC method are necessary. Information such as eluent composition, flow rate, column length, column temperature etc should be reported. For the quotation of detection limit in uM (e.g. page 7, line 14), the injection volume also matters. As this paper intends to present an improved IC method for HMS quantification, general aspects for evaluating and QA/QC an analytical method, such as calibrations curve, method precision (through repetitive analysis) and accuracy (through e.g. recovery analysis), and method robustness are important to be presented and discussed. It is also necessary to discuss the limitations and potential artifacts with the improved IC methods more thoroughly. For example, the stability of HMS is pH dependent. Loss of HMS is more severe at higher pH. The pH of the IC mobile phase is basic for eluent using sodium carbonate. The effect of HMS destruction during IC separating should be evaluated. Other issues such as the stability of the standard solution and potential loss of HMS in samples during storage are also important to discuss.

Minor comments:

Page 2, line 2, $HSO_3^-$ can dissociate at pH < 6.

Page 3 , line 12, define RSMS and ATOFMS.

Page 7, line 6, what is 4x200nm corresponding to?

---

## Author Comment (AC2) · 29 Jul 2019

We would like to thank Referee #3 for the comments that have helped improve the manuscript. The reviewer comments are in italic followed by our replies in normal text.

Comment 1 ("Title"): The title as it stands now is very broad. Maybe it could be phrased a bit more specfific ? Wouldn't it make sense to clearly mention HMS ?

The title has been revised to: "Measurement techniques of identifying and quantifying Hydroxymethanesulfonate in cloud water and particulate matter"

Comment 2 ("Introduction"): It covers quite some aspects, but at times there could be some more coverage. Maybe the authors can check again, HMS has been discussed a bit more often.

Information regarding the formation, chemistry and field measurements of HMS is presented throughout the introduction. As HMS is an important compound discussed in this work we provide information in all the paragraphs of the introduction.

Comment 3 ("Page 3 Section 1.2"): I find it starngethat here the very successlfully applied CE (capillary electrophoretic) separation and determination is not described. This is a mayor flaw and needs to be corrected. See Scheinhardt et al ., but especially references therein, Kramberger et al.

We have added a description of the CE method in page 4 lines 34-40: "Scheinhardt et al. (2014) provided evidence of identification of HMS during two field campaigns conducted in nine sites in Germany. Capillary electrophoresis (CE) was used resulting in efficient separation of HMS from other compounds when a voltage of -30 kV followed by hydrodynamic sample injection with 750 mbars was applied. Quantification was achieved through indirect UV detection at 260 nm wavelength and time resolution of 20 Hz. The detection limit of HMS was reported equal to 6-7 ng·m^(-3) and higher concentrations were observed during winter time. The method resulted successful quantification of HMS in concentration $\geq$18-21 ng·m^(-3). Concentrations in the range of 6-18 ng·m^(-3) were reported, however this range was characterized as less reliable in the study. (Scheinhardt et al., 2014)"

Comment 4 ("Page 5 line 29"): For MSA you should possibly reference Huang, Shan, et al. "Latitudinal and seasonal distribution of particulate MSA over the Atlantic using a validated quantification method with HR-ToF-AMS." Environmental science & technology 51.1 (2016): 418-426.

The citation of the recommended work has been added to the revised manuscript in page 6 line 11: "(Phinney et al., 2006; Huang et al., 2016; Chen et al., 2019)".

Comment 5 ("Page 6 Section 3.1"): Maybe it would be good to carry the conclusion of this section into the abstract: It is very difficult if not even impossible to identify or even quantify HMS through AMS only.

The conclusion of Section 3.1 is presented in the abstract in page 1 lines 18-22: "In cases where the dominant sulfur-containing species are ammonium sulfate or HMS, differences in AMS fragmentation patterns can be used to identify HMS. However, the AMS quantification of HMS in complex ambient mixtures containing multiple inorganic and organic sulfur species is challenging due to the lack of unique organic fragments and variability of fractional contributions of $H_x SO_y^+$ ions as a function of matrix."

Comment 6 ("HPLC"): Also, the HPLC method presented here does not fully convince. Please give numbers of merit for it and compare to all existing offline analytical techniques. Could you discuss wether AMS paralleled by filter sampling and CE analysis wouldn't be a valuable option? In this view, the discussion at the end of the paper should be widened.

Information and concentration ranges according to the study of Zuo and Chen (2003) are presented in page 4 lines 33-39. The study provides evidence of separation and quantification of HMS, sulfate and sulfite and the reported numbers that are relevant to the separation of these species are included in the manuscript. The present work does not aim to provide a literature review of the techniques that have been used to identify HMS thus Section 1.2 serves as a short discussion of methods previously used. The AMS coupled with CE analysis is an interesting option however since CE is not used in the present work, we could not comment on the efficiency of such a method. According to our finding AMS identification quantification of HMS is challenging. As pointed out by the referee #3, CE has successfully been used for the identification and quantification of HMS, however it is uncertain that the combination of the two system, AMS and CE, would result in better results.

---

## Author Comment (AC3) · 29 Jul 2019

We would like to thank Referee #1 for the comments that have helped improve the manuscript. The reviewer comments are in italic followed by our replies in normal text.

Major comments:

Comment 1 ("Title"): First of all, the title of the paper needs to be revised to better represent the main contents of the manuscript. For example, this work focuses exclusively on the AMS and IC methods, although there are other techniques available for HMS quantification. Also, the current title reads as if the method targets a range of sulfur containing compounds, but in reality it is primarily for HMS.

The title has been revised to: "Measurement techniques of identifying and quantifying

Hydroxymethanesulfonate in cloud water and particulate matter"

Comment 2 ("IC method information"): Secondary, more analytical details on the IC method are necessary. Information such as eluent composition, flow rate, column length, column temperature etc should be reported. For the quotation of detection limit in uM (e.g. page 7, line 14), the injection volume also matters. As this paper intends to present an improved IC method for HMS quantification, general aspects for evaluating and QA/QC an analytical method, such as calibrations curve, method precision (through repetitive analysis) and accuracy (through e.g. recovery analysis), and method robustness are important to be presented and discussed. It is also necessary to discuss the limitations and potential artifacts with the improved IC methods more thoroughly. For example, the stability of HMS is pH dependent. Loss of HMS is more severe at higher pH. The pH of the IC mobile phase is basic for eluent using sodium carbonate. The effect of HMS destruction during IC separating should be evaluated. Other issues such as the stability of the standard solution and potential loss of HMS in samples during storage are also important to discuss.

The eluent composition and flow rate are presented in Section 2.2.2 page 6 lines 1-2 of the revised manuscript. We have added the column and compartment temperature, delivery speed and delivery sample volume on page 6 lines 2-3: "The column and compartment temperatures were both 25oC and the delivery speed and delivery sample volume for the analysis were 4 mL/min and 4 mL." Information regarding the calibration curves, detection limit, accuracy, precision and robustness of our method have been added in the revised manuscript on page 7 and 8 lines 37-39 and 1-5, respectively: "The detection limits were determined by conducting sample runs of different concentrations. The concentration, C, for which the IC could not provide a clear peak was identified and samples runs were conducted for concentrations C+n, where n=0.2 mÎIJ. The concentration for which the baseline and the peak were clearly distinguishable was defined and 6 runs were conducted for this specific concentration to verify it. The uncertainty was determined, <1%, considering 99% confidence interval therefore

it was concluded that for the system used in this work the lowest corresponding concentration, for which a measurable peak was efficiently detected, is the detection limit. Standards were prepared before each experiment to ensure their stability and avoid possible decomposition if stored for a prolonged period of time.", page 7 lines 23-24: "Each sample analysis was conducted 4 times with individual sample preparation before each analysis. The area of the peaks was almost identical for sulfate and HMS in all 4 runs, with a difference only of 0.06 and 0.08 mM, respectively.", page 9 lines 3-8: "The eluent is also a technical aspect that differs between the two columns. The AS12A is an anion carbonate column, thus the eluent is neutral with respect to the pH, whereas the AS22 column is an anion hydroxide column, thus the eluent is basic with respect to pH. The stability of HMS has a strong pH dependence as it dissociates at high pH. The use of a neutral pH eluent avoids HMS decomposition during analysis. The majority of columns with alkyl quaternary ammonium functional group require neutral pH eluent, which also results in efficient separation of sulfur species." The pH of the eluent used for the AS12A was measured $\sim$7. The limitations of the method are presented in page 7 lines 30-32: "HMS and bisulfite/sulfite were not able to be separated as they had the same retention time in this case as well (Fig. 4). The efficiency and the clear separation of the peaks that the column provides allows for quantification of HMS when bisulfite/sulfite are not present." and page 8 lines 20-24: "If the concentrations are at lower levels, corresponding to <=30 $\mu$M of HMS, value experimentally estimated under laboratory conditions, which is equivalent to <=2 $\mu$g·m^(-3), assuming filter collection of ambient samples with sampling rate of $\sim$80 L·min, sampling time of $\sim$6 hr and extraction volume of 20 mL, an aliquot of which, 4mL, is used for sample analysis, and sulfate is of equal or higher concentration, the peaks corresponding to HMS and sulfate have lower area signals and will be treated as one peak. For pH=12 the peaks could not be distinguished."

Minor comments:

Comment 1 ("Page 2 line 2"): HSO3- can dissociate at pH < 6.

We have clarified our statement on the revised manuscript in page 2 line 3: "In cloud and fog water, $SO_2$ reacts with water producing bisulfite ($HSO_3^-$), when 3<pH<6, which further dissociates to form sulfite ($SO_3^{2-}$) when pH>6."

Comment 2 ("Page 3 line 12"): define RSMS and ATOFMS.

We have defined the acronyms RSMS and ATOFMS in the revised manuscript in page 3 lines 15-16: "(rapid single-particle mass spectrometer: RSMS, aerosol time-of-flight mass spectrometer: ATOFMS)"

Comment 3 ("Page 7 line 6"): what is 4x200nm corresponding to?

The statement 4x200 nm has a typo, it should be mm, and corresponds to diameter and length of the column. It has been corrected in all parts of the revised manuscript that it is mentioned. Page 7 lines 14 and 27: "(diameter=4 mm and length=250 mm of the column)" and "(diameter=4 mm and length=200 mm of the column)".

---

## Author Comment (AC4) · 29 Jul 2019

We would like to thank the Referee #2 for the comments that have helped improve the manuscript. The reviewer comments are in italics followed by our replies in normal text.

It is unfortunate that the authors decided not to test the AG18-AS18 columns used in the URG AIM-IC. I understand that testing a new column would require additional laboratory work, but I really believe it would significantly extend the usefulness of the manuscript, as the AIM-IC is being used by many researchers in China. In the reviewer response, the authors note that they expect that the columns "will not allow efficient separation of HMS and sulfate", but this is highly qualitative and likely depends on the eluent and run conditions. I urge the authors to reconsider adding the AG18-AS18

column testing to their work here.

Regarding the use of the AG18-AS18 columns: The separation efficiency of liquid chromatography columns is largely based on their functional groups. The conditions, temperature, sample volume and flow rate of the AG18-AS18 columns are the same as the AG22-AS22 columns according to technical specification of the AS18 column found in the manufacturer's website. Therefore, they will not affect the efficiency. Based on the functional groups of the AG18-AS18 columns we do not expect efficient separation due to the hydrophobicity of the analytical column and its functional group. In addition, the common eluent used is KOH. We chose the use of columns that require neutral eluent to avoid possible decomposition of HMS during the analysis which can be rapid at elevated pH.

Major comments:

Comment 1 ("Title"): The goal of this work, as stated through the paper, is to examine methods for the measurement of HMS in PM. Therefore, I suggest that the authors revise the title of their manuscript to specifically mention HMS and PM, rather than fog and cloud water.

The title has been revised to: "Measurement techniques of identifying and quantifying Hydroxymethanesulfonate in cloud water and particulate matter"

Comment 2 ("Page 7, Line 19 and Page 8, Lines 5-6"): The authors state "this method may result in noisy spectra for concentrations below 1 ppb" when discussing ESI-MS, but the paper cited (Chapman et al.) is from 1990. The signal/noise will depend on the mass analyzer used, in addition to the ionization method, and there have been great advances in mass analyzers and associated sensitivities over the past 30 years. Similarly, the LOD for the ESI-MS method is quoted as _100 ug/m3, but again, I expect this would have changed significantly over the time since publication. Therefore, these statements should be qualified, and rather future work should be motivated here to examine current sensitivities on ESI-MS instruments.

We agree with the referee on the comment that there have been important advances in mass analyzers, and it is possible that the concentration and LOD mentioned might be improved. However, the Chapman et al. (1990) paper is, to our knowledge, the main study that describes the use of ESI-MS for the identification and quantification of HMS. Therefore, we believe that reporting lower LODs from more recent studies that do not consider HMS will not be accurate for the purpose of this study. We have included a statement emphasizing that due to improvements in the instrumentation over time such LODs might be improved.

The statement is in page 3 lines 29-33 on the revised manuscript: "Chapman et al. (1990) conducted an exploratory study reporting that the quantitative detection limit for HMS can be in the order of 100 $\mu$g· m ˆ(-3) , for typical sampling conditions, using an ESI-MS. Since 1990 there have been advances in the ESI-MS technology that could possibly result in lower detection limits. However, to our knowledge, these technological changes have not yet provided quantitative evidence of lower detection limits with respect to HMS analysis."

Comment 3 ("Page 8, Line 11-12"): In reviewing Whiteaker et al (2003, Atmos. Environ.) based on the reviewer response, this paper does not cite a lack of sensitivity by the ATOFMS for detecting HMS, making the statement on Line 11 misleading. Rather, Whiteaker et al. discuss the matrix effects of ammonium and sodium, which impact the peak area detected; I cannot find evidence in this manuscript that the LOD for HMS would be high, indicating a lack of sensitivity, and no comparison is provided to other techniques. Therefore, I suggest the authors remove the phrase "and lack of sensitivity" and instead suggest in the paper that a study of the sensitivity of single-particle mass spectrometry instruments to HMS is an area of future work needed (as eluded to now on Page 7,Lines 37-38).

We would like to thank the referee #2 for the comment. The sentence aims to provide a general statement for single-particle mass spectrometry and along with the work of Whiteaker and Prather (2003), the studies of Neubauer at al. (1996 and 1997) are

cited. In the work of Neubauer et al. (1996 and 1997) it is stated that the assigned ion peak for HMS is "only observed from particles that contain a strong acid or proton donor" and that particles of a specific range were able to be examined. We have remove the statement "and lack of sensitivity" as we agree with the referee that these limitations can fall within the area matrix effects. We also agree on the necessity of stating that the sensitivity of these instruments with respect to HMS requires more detailed study in the future and we have included that statement on the revised manuscript.

The statement is in page 4 lines 26-27: "The sensitivity challenges of these methods with respect to HMS quantification yield the necessity of further study."

Comment 4 ("Page 7, Line 6-8"): PALMS is a single-particle mass spectrometry instrument. Please correct here. Please also note that the single-particle mass spec papers listed here do not represent a complete list, as implied. Either include "e.g." in front of the literature list, or conduct a more thorough literature search. Similarly, Section 1.2 describes each of the single-particle mass spectrometry studies listed here, but again, this is only a subset of published work on the subject, which is not reflected in the summaries presented. I'd encourage the authors to consider in Section 1.2 to conduct a more thorough literature search, and rather than describing each paper one-by-one, include a brief overview/summary of the observations.

We have corrected the phrase by including "single-particle". We included the single-particle mass spectrometry papers that have been used for identification and/or quantification of HMS. We clarify that in the manuscript in the relevant section which is in page 3 line 11: "A variety of technical methods have been used to detect HMS,…".

Comment 5 ("Page 11, Line 12-14"): The addition of the IC LODs and explanation of conversion to ambient mass concentration is very useful. However, please clarify how the LODs were determined and what they refer to, as there as multiple methods and definitions used in chromatography for LODs.

The LODs were determined by conducting sample runs of different concentrations.

The concentration, C, for which the IC could not provide a clear peak was identified and samples runs were conducted for concentrations C+n, where n=0.2 mM. The concentration for which the baseline and the peak were clearly distinguishable was defined and 6 runs were conducted for this specific concentration to verify it. We wanted 99% confidence interval therefore we calculated the standard deviation to also determine the uncertainty. The uncertainty was very low <1% therefore we concluded that for our system the lowest corresponding concentration, for which a measurable peak was efficiently detected, is the LOD. We have added this information in pages 7 and 8 lines 38-39 and 1-3, respectively: "The detection limits were determined by conducting sample runs of different concentrations. The concentration, C, for which the IC could not provide a clear peak was identified and samples runs were conducted for concentrations C+n, where n=0.2 mM. The concentration for which the baseline and the peak were clearly distinguishable was defined and 6 runs were conducted for this specific concentration to verify it. The uncertainty was determined, <1%, considering 99% confidence interval therefore it was concluded that for the system used in this work the lowest corresponding concentration, for which a measurable peak was efficiently detected, is the detection limit."

Comment 6 ("Page 11, Line 24-27"): I am confused at why a significant underestimation occurred when the elevated baseline was used. How was this determined? Was a calibration curve obtained and then a known concentration run to check? What is "significant" in this case? Please clarify.

We have clarified this information in the manuscript in page 8 and lines 17-20: "When this was applied a significant underestimation of the concentration, >=15% of HMS with 4% uncertainty, of the compounds was observed, therefore the software automatic separation was selected to be used. The percentages of HMS and sulfate were obtained considering the software separation of the peaks and the underestimation was determined by obtaining the calibration curves for sulfate and HMS and examining known concentrations.". A calibration curve was obtained for all the examined compounds and

a variety of know sample concentrations were tested. In all the cases we concluded that there was an underestimation of up to 15% of HMS, with 4% uncertainty, when the AS22 column was used, which is a significant underestimation.

Comment 7 ("Section 3.2"): What are the uncertainties in the percentages reported? Is reporting to one decimal place appropriate? Where the samples run in triplicate?

Each analysis was conducted 4 times with individual sample preparation before each analysis. For example, when we examined 2mM of HMS and 2mM of sulfate we prepared 4 different samples and analyzed them with IC. The area of the peaks was almost identical for sulfate and HMS in all 4 runs, with a difference only of 0.06 and 0.08 mM, respectively. Therefore, we concluded that it is accurate to report one decimal point.

Comment 8 ("Section 4"): Add a statement of the required concentration needed to distinguish HMS and sulfate (discussed on page 11), as this seems like it will significantly impact the recommendation of the necessary mass loading for ambient samples.

We have added that the required concentration needed to distinguish HMS and sulfate under the described conditions is >2 $\mu$g·mˆ(-3) of HMS and that sulfate concentration has to be lower than HMS. The statement is in page 9 lines 25-26: "Using an IC system, the detection limit of quantifying HMS and sulfate is 0.8 $\mu$M and 0.2 $\mu$M, respectively, and the required concentration needed to distinguish HMS and sulfate was determined to be >2 $\mu$g·mˆ(-3) of HMS and the sulfate concentration has to be lower concentration than that of HMS.".

Additional comments:

Comment 1 ("Section 1.1"): While the pivotal work of Munger et al 1986 (Science) is cited later in the manuscript, it would be highly valuable and most appropriate for this to be cited in the first paragraph of Section 1.1, as it sets the stage for the entirety of this work.

In the revised manuscript we now cite Munger et al. (1986) in the first paragraph of the
section where the HMS formation is mentioned. Page 1 line 29: "Hydroxymethanesulfonate (HMS; HOCH$_2$SO$_3^-$) is the product of the aqueous-phase reaction between dissolved sulfur dioxide (SO$_2$) and formaldehyde (HCHO) and is considered an important compound in cloud and fog water (Munger et al., 1986; Dixon and Aasen, 1999; Whiteaker and Prather, 2003).".

Comment 2 ("Page 2, Line 16-20"): Please provide references for these statements.

These statements refer to a response to a previous comment by the reviewer: "Metrohm MARGA uses a polystyrene/divinylbenzene copolymer with quaternary ammonium groups as functional group for separation of sulfite, sulfate and thiosulfate. Due to the fact that we are using the Dionex IC-5000+ IC model with adjustments in order to have a good separation of HMS and sulfate, compatibility issues might be encountered if we use a Metrohm column. It is possible that if the column used has polystyrene/divinylbenzene copolymer with alkyl quaternary ammonium group as functional group the separation of HMS and sulfate can be achieved, however we can not confidently make that statement. Unfortunately, we do not have access to this system for evaluation.". The statements results of discussion with technicians from Thermo Scientific, technical documents provided from Metrohm and experimental observations we obtained from ongoing projects.

Comment 3 ("Page 6, Line 27-28"): Rephrase statement "…measurements of HMS have mainly been conducted of fog and cloud water only" as there have been many ambient PM measurements of HMS by single-particle mass spectrometry.

We have rephrased the sentence according to the reviewer's recommendation. Page 2 lines 31-33: "Measurement of sulfate in ambient PM is common, whereas measurements of HMS have mainly been conducted for fog and cloud water. Studies reporting the presence of HMS in ambient PM using single-particle mass spectrometry have also been conducted (Neubauer et al., 1996; Neubauer et al., 1997; Whiteaker and Prather, 2003; Lee et al., 2003; Dall'Osto et al., 2009).".

[Figure]

Comment 4 ("Page 6, Line 30-32"): This statement is confusing as written. Please clarify.

We have clarified the statement in the revised manuscript. Page 2 lines 36-38: "Moreover, for MS, cations can be observed simultaneously in addition to sulfur-containing ions, whereas for IC a specified IC column with high sensitivity for sulfur-containing ions has to be used to identify them."

All acronyms have been specified in the revised manuscript.

Comment 5 and 6 ("Page 7, Line 29-30" and "Page 7, Line 35"): Single-particle mass spectrometers have several lasers. Please correct "operating laser" to "desorption/ionization laser". Matrix effects are inherent to the laser desorption/ionization process and have nothing to do with the inlet design, pump configuration, and reflectron. Rather matrix effects are associated with the competition in ion formation. Please correct here.

We have revised according to the reviewer's recommendation. Page 4 line 4 and line 10: "desorption/ionization laser at 266 nm" and "have been optimized to overcome sensitivity issues by improving the inlet design".

Comment 7 ("Section 1.2"): This section should describe the previous work of Gilardoni et al (2016, PNAS), who showed the AMS mass spectrum of HMS only. I realize that this paper is cited, but it would be useful for it to be described in the introduction to set the stage for how the current work builds upon this previous work.

We have included more information of the work of Gilardoni et al (2016) in the section 1.2. Page 4 lines 8-9: "Gilardoni et al. (2016) provided the spectrum of HMS using standard samples. During that study HMS was used as a tracer of aqueous chemistry.".

Comment 8 ("Page 7, Line 13"): Consider replacing "RSMS, PALMS, AToFMS" with "Single-particle Mass Spectrometry" as these are simply three of many types of single-particle mass spectrometers.

We have revised according to the reviewer's recommendation. Page 3 line 21: "1.2 Previous work identifying HMS using Single-particle Mass Spectrometry, Capillary Electrophoresis and reverse-phase HPLC".

Comment 10 ("Page 7, Line 29"): Correct "AToFMS" to "ATOFMS".

We have changed the "AToFMS" to "ATOFMS" in the manuscript.

Comment 11 ("Page 8, Line 5-6"): Please move these sentences the first paragraph of Section 1.2, where ESI-MS is discussed.

We have moved these sentences as suggested in page 3 lines 29-33 in the revised manuscript.

Comment 12 ("Page 8, Line 6-7"): Please clarify this sentence. Song et al. (2018) did detect HMS by SPAMS, even though the opposite seems to be stated in this sentence, with the opposite statement then in the following sentence.

We have clarified this sentence in the revised manuscript. The study by Song et al. 2019 (published) stated that the detection limit using AMS and SPMS could be lower than the detection limit reported by Chapman et al. (1990). In addition, the authors state that the SPMS data revealed that approximately 10% of HMS-containing particles in the total particles counts during haze events but they could not provide a quantitative measure particle as HMS, possibly due to fragmentation. Therefore, even though in that study HMS was able to be identified and it is stated that the detection limit could be lower than reported in the past, no quantitative information could be retrieved. Page 4 lines 20-21: "Although it was stated that the detection limit could possibly be lower using AMS and SPMS (Song et al., 2019) than the concentration reported by Chapman et al. (1990), 100 $\mu$g· m ^(-3) using ESI-MS„ such lower levels of HMS were not able to be detected using these methods. In their study, Song at al. (2019) were able to identify HMS as a component of SOA but they could not quantify it, likely for the reasons outlined below in this work.".

Comment 13 ("Page 9, Line 17"): Can you provide references here for the common use of this column?

The main reference is the technical report of the column provided by the manufacturer. The specific column, based on our experience, and the information provided by the manufacturer is the most common column for inorganic analysis.

Comment 14 ("Page 10, Line 7"): Please clarify "other species" here.

The phrase "the other species" refers to the other sulfur-containing compounds presented in Figure 1; sodium bisulfite, sodium sulfate and ammonium sulfate. We have clarified in the revised manuscript in page 6 lines 26-27 in the revised manuscript: "other species (sodium bisulfite, sodium sulfate and ammonium sulfate)".

Comment 15 ("Page 10, Line 14"): Fix reference formatting here.

The reference formatting has been corrected in the revised manuscript.

Comment 16 ("Section 3.2"): Some of this section repeats the methods and could be condensed.

In the experimental section general information is provided however, in the section 3.2 we present more detailed information for each examined column pair.

Comment 17 ("Page 13, Line 1"): I'm confused by the statement "Applications of both IC and AMS methods to the same ambient samples in the future" as isn't a finding of this work that the AMS is unable to distinguish between HMS and sulfate. Also, I'm confused because I thought these samples were not available for analysis based on the reviewer response. Please clarify.

We have clarified this sentence as our intent was to point out that it would be useful to use the methods we describe in this work to analyze the ambient samples, or similar samples from severe haze events and specifically samples from similar conditions of the work of Wang et al. (2014). Page 10 line 1 in the revised manuscript: "Applications
of both IC and AMS methods to the same ambient samples from similar conditions of the January 2013 haze event".

---

## Author Response (AR2)

**Response to report #2 and resubmission of the manuscript "amt-2019-127"**

We would like to thank Referee #2 for the comments. The reviewer comments are in italic followed by our replies in normal text.

**Major comments:**

Comment 1: *Title (all three reviewers): The title revision to mention HMS and PM is an improvement. However, it is misleading to mention cloud water, since this was not measured in this study. I also highly recommend specifically mentioning AMS and IC in the title so that the manuscript can be more easily identified by researchers searching on these methods. The other techniques discussed in the manuscript are simply in the background/intro material and do not constitute new knowledge added through this*

*study.*

The title has been revised to: "Measurement techniques of identifying and quantifying hydroxymethanesulfonate (HMS) in aqueous matrix and particulate matter using aerosol mass spectrometry and ion chromatography"

Comment 2: *Limit of Detection (LOD), Pages 7-8 (Response to Reviewer 1, Comment 2 & Reviewer 2, Comment 5): I am concerned by how the authors have defined and determined their detection limit by running multiple concentrations and defining the LOD as the lowest concentration for which a measurable peak was efficiently quantified. Under most circumstances, LOD is defined at the*

*3\* standard deviation of the noise +/- uncertainty and is not determined in the way described by the authors. I believe the Thermo Chromeleon software provides a LOD based off the Hubaux and Vos method using an entire calibration curve. However, Ragland et al. (2014, Analytical Chem.) described the weaknesses of this method for chromatography, and even the Hubaux and Vos method was not used by the authors for an unknown reason. A recommended method often used to calculate the LOD from a calibration curve is to calculate 3\*sensitivity/slope. Especially since this is a method paper, it is important that the LODs be reported*

*accurately.*

We understand that the way we described the LOD determination might cause confusion, thus we have corrected the statement to include more detailed information on page 8 and line 7-16: "The detection limits were determined by conducting sample runs of different concentrations. The concentration, C, for which the IC could not provide a clear peak was identified and sample runs were conducted for concentrations C+n, where n=0.2 mM. The concentration for which the baseline and the peak were clearly distinguishable was defined and 6 runs were conducted for this specific concentration to verify it. The standard deviation of that concentration was estimated. Blank samples were analysed for each compound and the mean value and the standard deviation was determined. Considering 99% confidence interval the limit of blank was calculated as the mean blank value plus the product of the standard deviation of the blank and 2.58, value which corresponds to 99% confidence level (Limit of blank = (mean of bank) +

2.58·(standard deviation of blank)). The detection limit was estimated as the sum of the limit of blank and the product of the standard deviation of the low concentration and 2.58 (Detection limit = (Limit of bank) + 2.58·(standard deviation of low concentration)).".

The referee is correct on the fact that we did not use the definition of "3* standard deviation of the noise +/- uncertainty" or the "3*sensitivity/slope". The method used was the one including the limit of blank described also by Shrivastrava and Gupta, 2011 and Armbruster and Pry, 2008. As described in the manuscript we determined the lower concentration in which the peak was efficiently detected having 99% confident interval. To be more specific, once a range of concentrations in which we show an efficient signal was established we calculated the mean value and the standard deviation of blank samples. The limit of blank was then calculated as the mean value plus (2.58*standard deviation of the blank). Then, standard deviation of the lowest concentration was determined and the LOD was estimated as the limit of blank plus (2.58*standard deviation of the lowest concentration). The 2.58 provides the 99% confidence interval.

The equations describing the aforementioned information are:

$$\text{Limit of blank} = (\text{mean of bank}) + 2.58 \cdot (\text{standard deviation of blank})$$

$$\text{Detection limit} = (\text{Limit of bank}) + 2.58 \cdot (\text{standard deviation of low concentration})$$

We have also calculated the LODs according the method proposed by the referee, 3*sensitivity/slope, and the results that we obtain are 0.16 µM and 0.6 µM for sulfate and HMS, respectively. Our approach is more conservative as we include two calculations of standard deviation.

Comment 3: *Uncertainty (Reviewer 2, Comments 6-7): Related to the last comment, how can there only be a 4% uncertainty when there is an underestimation of over 15% of HMS? Did this occur near the detection limit, such that this may be an issue of being below the limit of quantitation (typically defined at 3.3*LOD)? The points here about limit of detection are also related to the determination of uncertainty. Typically uncertainty is reported as 3*standard deviation of triplicate runs, rather than by just looking at the differences in the values and choosing one decimal point (response to Reviewer 2, Comment 7); however, again, the authors did not assess this rigorously, which again opens questions about the analytical methodology in this manuscript.*

The two values of uncertainty and underestimation provide information regarding the precision and the accuracy, respectively. The 4% uncertainty is the precision of the HMS concentration measurements. To be more specific, samples containing the same ratio of [HMS]/[sulfate] were prepared and analyzed. Since these samples had the same HMS concentration the standard deviation of the areas of the peaks corresponding to HMS was calculated. The obtained value is the precision and is in area units thus it is converted using the calibration curve to concentration units.

The 15% underestimation is calculated considering samples that contained both HMS and sulfate and samples that contained only HMS in the same concentration as the samples with HMS and sulfate. Therefore, the percentage provides information of the accuracy of the measurement when these two types of samples are analyzed. For example, a sample that contained C1 concentration of HMS had A1 area corresponding to the HMS peak. A sample containing C1 concentration of HMS and C2 concentration of sulfate had A2 area corresponding to HMS. Calculating the concentration according to the calibration curve we noticed that the concentration of HMS in the second case was not C1 but 15% lower when the AS22 column was used. Therefore, we present the issue of accuracy for the AS22 column and columns of the same type.

We have added a clarifying statement on page 8 lines 30-34: "To be more specific, 4% uncertainty corresponds to the concentrations measured in multiple runs, thus the precision of the HMS concentration measurements, and ≳15% underestimation is the underestimation in HMS concentration when a sample containing both sulfate and HMS is analysed and compared with the HMS concentration of a sample containing only HMS. Therefore, the percent underestimation shows the lack of accuracy of the measurements when these two sample types are analysed with the AS22 column."

We calculated the uncertainty according to the referee's recommendation, 3*standard deviation of triplicates runs, and the result is the same as following our method.

In a previous comment regarding the one decimal point "What are the uncertainties in the percentages reported? Is reporting to one decimal place appropriate? Where the samples run in triplicate?" we argued that the one decimal selection is due to the 2 decimal point difference that we observe in concentrations. The sample runs were conducted 4-6 times in order to verify the accuracy of the results that we are reporting.

Comment 4: *Reviewer 1, Comment 2; Page 9: The pH of the eluent is noted, but where is the actual IC eluent composition and concentration provided? Further, it is stated that the stability of the HMS has a strong pH dependence. What is the pKa of HMS? This would be useful to report.*

The eluent composition and concentration are mentioned in section 2.2.2 (pages 5 and 6, lines 36 and 1): "The mobile phase during the experiments was 4.5 mM:0.8 mM sodium carbonate: sodium bicarbonate with flow rate $1 \text{ mL} \cdot \text{min}^{-1}$.". The pKa value has been added to the revised manuscript on pages 1 and 2 lines 33-35 and 1, respectively: "The stability of HMS has a strong pH dependence as it dissociates at high pH values. HMS acid is a strong acid, thus it completely dissociates in water, with a second dissociation constant of pKa=10.2 (R1). (Olson and Hoffmann, 1986; Betterton et al., 1988; Olson and Hoffmann, 1989; Warneck, 1989; Möller, 2014)

$$HCHO + SO_3^{2-} \rightarrow HCH_{(O)}{}^{-}SO_3^{-} \overset{H^+}{\rightleftharpoons} HOCH_2SO_3^{-} \quad (R1)"$$

Comment 5: *Page 9, Lines 24-26: Please report the required concentration to distinguish sulfate and HMS in uM, rather than ug m-3, which is far less useful, since it depends on filter extraction protocol. It is stated that sulfate needs to be in lower concentration than HMS for the method to be viable, but is there a scenario in the atmosphere where this is expected to occur?*

The required concentration to distinguish HMS in μM has been added to page 10 line 4: "…the required concentration needed to distinguish HMS and sulfate was determined to be >30 μM of HMS…".

In conditions where HCHO is in high levels and peroxides are in lower levels, conditions that are observed during haze events, HMS can be in higher concentrations compared to sulfate.

Comment 6: *Reviewer 3, Comment 3: Based on comparison of the discussion of the method in the Conclusions section and throughout, it would appear that CE would be a superior technique in terms of efficient separation and sensitivity to detect HMS, based on Scheinhardt et al (2014), which has now been added to the manuscript. For example, the LOQ was determined to be ~20 ng m-3 in that study, compared to >2 ug m-3 in this study. If these direct comparisons are not fair, then this should be even further discussed, especially in the conclusions section when the reviewer is assessing the best methods going forward.*

Scheinhardt et al. (2014) reported the LOD to be lower compared to the LOD reported in our work, however the authors also characterized the quantification of HMS in concentrations close to the reported LOD as unreliable. This information is provided on the manuscript on page 4 lines 38-39. We agree that CE may very well be the best approach for detection of HMS, unfortunately, there is not much information in the literature on this. We have nonetheless added a statement "Dabek-Zlotorzynska et al. (2002 and 2005) analysed urban atmospheric aerosol and vehicle emitted samples using CE and reported the presence of HMS. Identification and quantification were achieved using two injection modes, pressure and electrokinetic injection modes, for CE and the detection limits of HMS reported were 0.4 μM and 0.02 μM, respectively. The wavelength of the UV detector for indirect detection was at 214 nm. (Dabek-Zlotorzynska et al., 2002) CE may be a method that is highly suitable for detection of HMS and it has high sensitivity and the eluent pH also should prevent decomposition of HMS." (page 5 lines 1-6).

**Minor comments:**

Comment 1: *Page 4, Lines 19-21: Please fix this sentence as it has typos and is not understandable as written.*

We have rephrased the sentence (page 4, lines 21-23): "Although Song et al (2019) stated that the detection limit of AMS and SPMS for HMS could possibly be lower than the concentration reported by Chapman et al. (1990), 100 $\mu g \cdot m^{-3}$ using ESI-MS, such lower levels of HMS were not able to be detected using these methods."

Comment 2: *Page 3, Lines 14-16: Fix the phrasing "single-particle analysis by laser mass spectrometry (PALMS) (Lee et al., 2003) and single-particle mass spectrometry (rapid…" to "single-particle mass spectrometry (single-particle analysis by laser mass spectrometry: PALMS, Lee et al., 2003; rapid…" since PALMS is a single-particle mass spectrometry technique.*

[revised manuscript text omitted]